# Stability and Sharper Risk Bounds with Convergence Rate $\tilde{O}(1/n^2)$

**Bowei Zhu**[1,2,3], **Shaojie Li**[1,2,3,*], **Mingyang Yi**[1,*], **Yong Liu**[1,2,3,*]

[1]Gaoling School of Artificial Intelligence, Renmin University of China, Beijing, China
[2]Beijing Key Laboratory of Research on Large Models and Intelligent Governance
[3]Engineering Research Center of Next-Generation Intelligent Search and Recommendation, MOE
{bowei.zhu, lishaojie95, yimingyang, liuyonggsai}@ruc.edu.cn

## Abstract

Prior work (Klochkov & Zhivotovskiy, 2021) establishes at most $O\left(\log(n)/n\right)$ excess risk bounds via algorithmic stability for strongly-convex learners with high probability. We show that under the similar common assumptions — Polyak-Lojasiewicz condition, smoothness, and Lipschitz continous for losses — rates of $O\left(\log^2(n)/n^2\right)$ are at most achievable. To our knowledge, our analysis also provides the tightest high-probability bounds for gradient-based generalization gaps in nonconvex settings.

## 1 Introduction

Algorithmic stability is a fundamental concept in learning theory [3], which can be traced back to the foundational works of Vapnik and Chervonenkis [46] and has a deep connection with learnability [37, 40, 39]. Roughly speaking, an algorithm is stable if a substitution of single example in the training dataset leads to a minor change of the output model. With such algorithmic stability, the generalization ability of model is guaranteed. Owing to the relationship, the algorithmic stability is recognized as a powerful tool to explore the generalization.

In practice, researchers built the generalization bounds expressed in-expectation or in-high probability. While providing in-expectation bounds is relatively straightforward, their high probability counterparts are more crucial for practical optimization algorithms [12, 4, 20]. Because we often train models single time in practice, and high probability bounds offer more informative insights. Therefore, we focus on improving high probability risk bounds through an exploration of algorithmic stability.

Exploring the high-probability generalization gap bounded via algorithmic stability was firstly chased to Bousquet and Elisseeff [3], and was further developed by Feldman and Vondrak [11, 12]. Within these literature, the optimal high-probability bound of generalization error consists of the "sampling error" $O\left(\log(n)/\sqrt{n}\right)$ plus the algorithmic stability parameter for bounded loss function. Owing to the existence of sampling error, deriving the bound within the framework in [4] can not be improved, since the lower bound of sampling error is proven to be $O(1/\sqrt{n})$. To resolve this, Klochkov and Zhivotovskiy [20] propose to alternatively explore the **excess risk**, which directly reveals the performance of learned model, and can be decomposed into optimization and generalization errors. Without considering the sampling error, Klochkov and Zhivotovskiy [20] improve the high-probability bound to excess risk of $O(\log(n)/n)$ plus algorithmic stability parameter, under a Bernstein type condition [49]. Moreover, under strongly convexity condition, the algorithmic stability parameter of projected gradient descent (PGD) is $O\left(\log(n)/n\right)$ [20]. Following this, the algorithmic stability-

---

*Corresponding author.

based excess risk bounds were further explored by Yuan and Li [52, 53], Yi et al. [51], while their results are all weaker than $O(\log(n)/n)$. Thus, a natural question is:

*Can algorithmic stability-based technique provide high probability excess risk bounds better than $O(1/n)$?*

The main results of this paper answer this question positively. Roughly speaking, under proper regularity conditions, *we establish the first high probability excess risk bounds that are dimension-free with the rate $O\left(\log^2(n)/n^2\right)$* for models obtained by empirical risk minimization (ERM), PGD and stochastic gradient descent (SGD). To do so, our core technique is generalizing the standard concept of algorithmic stability of loss function to its gradient, i.e., the "uniform stability in gradients" in Definition 2 [24]. With this, we can connect it with a sharper generalization bound, compared with the classical ones [4].

Next, we outline our idea to the sharper bound to excess risk. Notably, under some regularity conditions e.g. strongly convexity or Polyak-Lojasiewicz (PL) condition [20], the loss function is upper bounded by the square of its gradient norm. Then, by invoking the algorithmic stability of gradient, we prove a *generalization bound w.r.t. gradient of loss function*. With this, an upper bound to the gradient of excess risk is obtained, which naturally leads to the bound of excess risk, under strongly convexity of PL conditions. During the proof, our generalization bound w.r.t. gradient consists of algorithmic stability coefficient, gradient of loss function on trained parameters, and terms in $O(1/n)$. Intuitively, the gradient related term on trained model tends to close zero. Thus, combining the derived generalization bound on gradient and analysis on optimization error induces our sharper bound $O\left(\log^2(n)/n^2\right)$ on excess risk.

Finally, we summarize our contributions as follows, and compare our results with the previous ones in Table 1.

- Under proper regularity conditions, we sharpen the algorithmic stability-based generalization bound i.e., from $\tilde{O}\left(1/n\right)$ [20] to $\tilde{O}\left(1/n^2\right)$ via the stability of gradients. To the best of our knowledge, this is optimal dimension-free results based on algorithmic stability up till now.
- With our framework, we derive the first dimension-free high probability excess risk bounds of $O\left(\log^2(n)/n^2\right)$ for ERM, PGD, and SGD, addressing an open problem posed in Xu and Zeevi [50]. For SGD, we have also shown that under the same assumptions, an equally tight bound can be achieved with fewer iterations.

**Notations.** In this paper, we consider a set of training data independent and identically distributed (i.i.d.) observations $S = \{z_1, \ldots, z_n\}$ sampled from a probability distribution $\rho$ defined on $\mathcal{Z}$, $S' = \{z'_1, \ldots, z'_n\}$ be its independent copy. For any $i \in [n]$, define $S^{(i)} = \{z_i, \ldots, z_{i-1}, z'_i, z_{i+1}, \ldots, z_n\}$ by replacing the $i$-th sample in $S$ with another i.i.d. sample $z'_i$. Based on the training set $S$, our goal is to build a model with parameter $\mathbf{w} \in \mathcal{W} \subset \mathbb{R}^d$. We denote the performance of model with parameters $\mathbf{w}$ evaluated on $z$ by loss function $f(\mathbf{w}; z)$, where $f : \mathcal{W} \times \mathcal{Z} \mapsto \mathbb{R}_+$. Then the population risk and the empirical risk of $\mathbf{w} \in \mathcal{W}$, are respectively denoted as

$$F(\mathbf{w}) := \mathbb{E}_{z \sim \rho}\left[f(\mathbf{w}; z)\right], \quad F_S(\mathbf{w}) := \frac{1}{n}\sum_{i=1}^{n} f(\mathbf{w}; z_i).$$

We respectively denote $\mathbf{w}^* \in \arg\min_{\mathbf{w} \in \mathcal{W}} F(\mathbf{w})$, and $\mathbf{w}^*_S \in \arg\min_{\mathbf{w} \in \mathcal{W}} F_S(\mathbf{w})$, and learned model parameters $A(S) \in \mathcal{W}$ be the output of a (possibly randomized) algorithm $A$ on the training set $S$. In general, we will care the generalization error of model $A(S)$ defined as the gap between population risk and empirical risk i.e., $F(A(S)) - F_S(A(S))$. Besides, the excess risk $F(A(S)) - F(\mathbf{w}^*)$ decides the performance of model is more important in practice. Finally, the notation $\tilde{O}(\cdot)$ hides logarithmic factors and we denote $L_p$-norm of real value $Y$ as $\|Y\|_p := (\mathbb{E}[|Y|^p])^{\frac{1}{p}}$. Similarly, let $\|\cdot\|$ denote the norm in a Hilbert space $\mathcal{H}$. For a random variable $X$ taking values in a Hilbert space, its $L_p$-norm is defined by $\||\mathbf{X}|\|_p := (\mathbb{E}[\|\mathbf{X}\|^p])^{\frac{1}{p}}$.

## 2 Related Work

Algorithmic stability is a classical approach in generalization analysis, which can be traced back to the foundational works of Vapnik and Chervonenkis [46]. It gave the generalization bound by

Table 1: Summary of high probability excess risk bounds. All conclusions herein assume Lipschitz continuity, and all SGD algorithms presuppose bounded variance of the gradients; therefore, these two assumptions are omitted in the table. Abbreviations: uniform convergence → UC, algorithmic stability → AS, strongly convex → SC, low noise [55] → LN, Polyak-Lojasiewicz condition → PL.

| Reference | Algorithm | Method | Assumptions | Iterations | Sample Size | Bounds |
|---|---|---|---|---|---|---|
| [55] | ERM | UC | Smooth, SC, LN | - | $\Omega\left(\frac{\gamma^2 d}{\mu^2}\right)$ | $\tilde{O}\left(\frac{1}{n^2}\right)$ |
| [50] | ERM | UC | Smooth, PL, LN | - | $\Omega\left(\frac{\gamma^2 d}{\mu^2}\right)$ | $\tilde{O}\left(\frac{1}{n^2}\right)$ |
| | PGD | UC | Smooth, PL, LN | $T \asymp \log(n)$ | $\Omega\left(\frac{\gamma^2 d}{\mu^2}\right)$ | $\tilde{O}\left(\frac{1}{n^2}\right)$ |
| [29] | SGD | UC | Smooth, PL, LN | $T \asymp n^2$ | $\Omega\left(\frac{\gamma^2 d}{\mu^2}\right)$ | $\tilde{O}\left(\frac{1}{n^2}\right)$ |
| | | AS | Smooth, SC | $T \asymp n^2$ | - | $\tilde{O}\left(\frac{1}{n}\right)$ |
| [20] | ERM | AS | SC | - | - | $\tilde{O}\left(\frac{1}{n}\right)$ |
| | PGD | AS | Smooth, SC | $T \asymp \log(n)$ | - | $\tilde{O}\left(\frac{1}{n}\right)$ |
| This work | ERM | AS | Smooth, PL, LN | - | $\Omega\left(\frac{\gamma^2}{\mu^2}\right)$ | $\tilde{O}\left(\frac{1}{n^2}\right)$ |
| | PGD | AS | Smooth, PL, LN | $T \asymp \log(n)$ | $\Omega\left(\frac{\gamma^2}{\mu^2}\right)$ | $\tilde{O}\left(\frac{1}{n^2}\right)$ |
| | PGD | AS | Smooth, PL, LN | $T \asymp \log(n)$ | $\Omega\left(\frac{\gamma^2}{\mu^2}\right)$ | $\tilde{O}\left(\frac{1}{n^2}\right)$ |
| | SGD | AS | Smooth, PL, LN | $T \asymp n^2$ | $\Omega\left(\frac{\gamma^2}{\mu^2}\right)$ | $\tilde{O}\left(\frac{1}{n^2}\right)$ |
| | SGD | AS | Smooth, PL | $T \asymp n$ | $\Omega\left(\frac{\gamma^2}{\mu^2}\right)$ | $\tilde{O}\left(\frac{1}{n}\right)$ |

analyzing the sensitivity of a particular learning algorithm when changing one data point in the dataset. Modern method of stability analysis was established by Bousquet and Elisseeff [3], where they presented the concept of uniform stability.

Since then, a lot of works based on uniform stability have emerged. Some of the existing results built their bound in expectation [17, 51, 5, 8], while it is weaker than the high probability bound as we discussed. To resolve this, Bousquet and Elisseeff [3], Elisseeff et al. [9], Feldman and Vondrak [11, 12], Bousquet et al. [4], Klochkov and Zhivotovskiy [20], Yuan and Li [52, 53], Fan and Lei [10] considered high probability bounds. Besides that, some other generalized algorithmic stability measures are further developed to study the generalization. For instances, uniform argument stability [31, 1], uniform stability in gradients [24, 10], on average stability [40, 21], hypothesis stability [3, 5], hypothesis set stability [13], pointwise uniform stability [10], PAC-Bayesian stability [28], locally elastic stability [8], and collective stability [32]. However, the optimal generalization bound among these results is $O(\log(n)/n)$, which is weaker than our $O\left(\log^2(n)/n^2\right)$ in this paper.

Notably, the $O\left(\log^2(n)/n^2\right)$ generalization bound under PL condition or strongly convexity has also been developed by uniform convergence technique as in Zhang et al. [55], Xu and Zeevi [50]. However, in contrast to ours, their results are restricted to the number of samples $n = \Omega(d)$.

## 3   Stability and Generalization

In this section, we first introduce the stability of gradient, and connect it to the generalization bound w.r.t. gradient. Following this, under proper regularity condition, i.e., PL condition, we can extrapolate the generalization of gradient to the loss function. With the desired generalization bound, by combining it with the optimization error, we can further bound the excess risk.

Firstly, let us introduce some basic definitions here. Before presenting our main results (Theorem 1 and Theorem 2) i.e., the generalization bound w.r.t. gradient, we emphasis that they do not require the smoothness assumption and PL condition. This indicates their potential applications within the nonconvex problems as well.

**Definition 1.** *Let* $f : \mathcal{W} \mapsto \mathbb{R}$. *Let* $M, \gamma, \mu > 0$.

- *We say $f$ is $M$-Lipschitz if*

$$|f(\mathbf{w}) - f(\mathbf{w}')| \leq M\|\mathbf{w} - \mathbf{w}'\|_2, \quad \forall \mathbf{w}, \mathbf{w}' \in \mathcal{W}.$$

- *We say $f$ is $\gamma$-smooth if*

$$\|\nabla f(\mathbf{w}) - \nabla f(\mathbf{w}')\|_2 \leq \gamma\|\mathbf{w} - \mathbf{w}'\|_2, \quad \forall \mathbf{w}, \mathbf{w}' \in \mathcal{W}.$$

- *Let $f^* = \min_{\mathbf{w} \in \mathcal{W}} f(\mathbf{w})$. We say $f$ satisfies the Polyak-Lojasiewicz (PL) condition with parameter $\mu > 0$ on $\mathcal{W}$ if*

$$f(\mathbf{w}) - f^* \leq \frac{1}{2\mu}\|\nabla f(\mathbf{w})\|_2^2, \quad \forall \mathbf{w} \in \mathcal{W}.$$

### 3.1 Sharper Generalization Bounds in Gradients

Let us start with the notation of algorithmic stability w.r.t. gradient.

**Definition 2** (Uniform Stability in Gradients). *Let $A$ be a randomized algorithm. We say $A$ is $\beta$-uniformly-stable in gradients if for all neighboring datasets $S, S^{(i)}$, we have*

$$\sup_z \mathbb{E}_A \left[ \left\| \nabla f(A(S); z) - \nabla f(A(S^{(i)}); z) \right\|_2^2 \right] \leq \beta^2. \tag{1}$$

**Remark 1.** The notation of gradient-based stability was firstly introduced by Lei [24], Fan and Lei [10] to describe the generalization performance for nonconvex problems. Because in this regime, the exploration usually focuses on the first-order critical condition i.e., $\|\nabla F(A(S))\|_2$. Then, with the stability on gradient, the "generalization of gradient" can be expressed $\|\nabla F(A(S)) - \nabla F_S(A(S))\|_2$. Thus, combining it with a triangle inequality and standard results on gradient of empirical risk $\|\nabla F_S(A(S))\|_2$ characterizes the desired $\|\nabla F(A(S))\|_2$. Further more, under a small $\|\nabla F(A(S))\|_2$, the PL condition (see Definition 1) leads to the desired bound on excess risk. This explains why the gradient-based stability is the core idea of this paper.

With the uniform stability in gradients, we prove that it implies the generalization of gradient in the following theorem.

**Theorem 1** (Generalization via Stability in Gradients [10]). *Assume for any $z$, $f(\cdot, z)$ is $M$-Lipschitz. If $A$ is $\beta$-uniformly-stable in gradients, then for any $\delta \in (0, 1)$, the following inequality holds with probability at least $1 - \delta$*

$$
\mathbb{E}_A \left[ \|\nabla F(A(S)) - \nabla F_S(A(S))\|_2 \right] \leq 2\beta + \frac{4M \left( 1 + e\sqrt{2\log(e/\delta)} \right)}{\sqrt{n}}
$$
$$
+ 8 \times 2^{\frac{1}{4}} (\sqrt{2} + 1)\sqrt{e}\beta \lceil \log_2(n) \rceil \log(e/\delta).
$$

**Remark 2.** Notably, our Theorem 1 is similar to Theorem 3 in [10], while ours has a constant-level improvement owing to a sharper concentration inequality i.e., Lemma 7 in Appendix. However, the generalization bounds in both of the aforementioned theorems have dependence $O\left(M/\sqrt{n}\right)$, which leads to a $O(M^2/n)$ generalization bound of loss function, and it is weaker than our desired result. Thus, we derive the following sharper generalization bound of gradients under same assumptions.

**Theorem 2** (Sharper Generalization via Stability in Gradients). *Assume for any $z$, $f(\cdot, z)$ is $M$-Lipschitz. If $A$ is $\beta$-uniformly-stable in gradients, then for any $\delta \in (0, 1)$, the following inequality holds with probability at least $1 - \delta$*

$$
\mathbb{E}_A \left[ \|\nabla F(A(S)) - \nabla F_S(A(S))\|_2 \right]
$$
$$
\leq \sqrt{\frac{4\mathbb{E}_{Z,A} \left[ \|\nabla f(A(S); Z)\|_2^2 \right] \log(6/\delta)}{n}} + \sqrt{\frac{\left( \frac{1}{2}\beta^2 + 32n\beta^2 \log(3/\delta) \right) \log(6/\delta)}{n}}
$$
$$
+ \frac{M \log(6/\delta)}{n} + 16 \times 2^{\frac{3}{4}} \sqrt{e}\beta \lceil \log_2(n) \rceil \log(3e/\delta) + 32\sqrt{e}\beta \lceil \log_2(n) \rceil \sqrt{\log(3e/\delta)}.
$$

**Remark 3.** We begin by comparing the generalization bound of gradient in the proposed Theorem 2 with Theorem 1. The critical difference is that constants in $1/\sqrt{n}$ is improved from $O\left(M\sqrt{\log(e/\delta)}\right)$ to $O\left(\sqrt{\mathbb{E}_Z \left[\|\nabla f(A(S); Z)\|_2^2\right] \log(1/\delta)} + \beta \log(1/\delta)\right)$, since $M$ is the maximal gradient norm. Notably, the proved dependence $\mathbb{E}_Z \left[\|\nabla f(A(S); Z)\|_2^2\right]$ is interpreted as the

squared gradient norm of the loss functions on test data, under the optimized model parameters $A(S)$. Thus, it is supposed to be small, compared with constant $M$, since the optimization algorithms often provide parameters approaching the optimal solution.

Moreover, our bound is not restricted to any specific algorithm and without dependence on the number of optimization iterations. Clearly, this is an improvement to the results constructed under a specific algorithm, e.g., the $O(T/n)$ generalization bound under SGD in Bassily et al. [1], Lei [24], which is only applied to SGD, and becomes vacuous when iteration steps $T$ exceeds $n$.

Next, we give the proof sketch to our Theorem 2, which is motivated by the analysis in Klochkov and Zhivotovskiy [20]. Similar to the standard result in Bousquet and Elisseeff [3], which says the generalization of loss function is implied by its stability and the variance of stability. Similar result is generalized to the gradient. During our proof, such algorithmic stability of gradient is characterized by the gap $\mathbf{q}_i(S) = \mathbf{h}_i(S) - \mathbb{E}_{S \setminus \{z_i\}, A}[\mathbf{h}_i(S)]$ with $\mathbf{h}_i(S) = \mathbb{E}_{z_i'} \left[ \mathbb{E}_Z \left[ \nabla f(A(S^{(i)}), Z) \right] - \nabla f(A(S^{(i)}), z_i) \right]$ and its variance. The stability is $\beta$ appears in Theorem 2. While instead of controlling its variance with maximal gradient norm, we can link it to the coefficient $\mathbb{E}_Z \left[ \|\nabla f(A(S); Z)\|_2^2 \right]$, which implies our result. We refer readers for more details to the proof in Appendix.

To further characterize the $\mathbb{E}_Z \left[ \|\nabla f(A(S); Z)\|_2^2 \right]$, we need the strong growth condition (SGC) imposed in Solodov [42], Vaswani et al. [48], Lei [24], which is satisfied many important loss function, e.g., squared-hinge loss with finite support [48].

**Definition 3** (Strong Growth Condition). *The SGC holds, if*

$$\mathbb{E}_Z \left[ \|\nabla f(\mathbf{w}; Z)\|_2^2 \right] \leq \lambda \|\nabla F(\mathbf{w})\|_2^2.$$

**Proposition 1** (SGC case). *Let assumptions in Theorem 2 hold and suppose SGC holds. Then for any $\delta > 0$, $\eta > 0$, with probability at least $1 - \delta$, we have*

$$\mathbb{E}_A \left[ \|\nabla F(A(S))\|_2 \right] \lesssim (1+\eta)\mathbb{E}_A \left[ \|\nabla F_S(A(S))\|_2 \right] + \frac{1+\eta}{\eta} \left( \frac{\lambda M \log(1/\delta)}{n} + \beta \log(n) \log(1/\delta) \right).$$

Proposition 1 build a connection between the population gradient error and the empirical ones under Lipschitz, nonconvex, nonsmooth and SGC conditions. The conclusion is directly obtained from the triangle inequality, Young's inequality, and SGC. Since the model parameters $A(S)$ is optimized, the $\|F_S(A(S))\|$ is supposed to be small. Thus, the Proposition 1 is a valuable upper bound on that of population error, for algorithm with stability w.r.t. gradient (small $\beta$).

**Remark 4.** Finally, we make a discussion to the priority of our Theorem 2, solely in that of gradient generalization. In a word, we address an open problem posed by [50], namely achieving a bound to generalization error of gradient independent of the dimension $d$. Concretely, Xu and Zeevi [50] prove a generalization bound via uniform convergence technique [47] is

$$\|\nabla F(A(S)) - \nabla F_S(A(S))\|_2$$
$$\lesssim \sqrt{\frac{\mathbb{E}_Z \left[ \|\nabla f(\mathbf{w}^*; Z)\|_2^2 \right] \log(1/\delta)}{n}} + \frac{\log(1/\delta)}{n} + \max \left\{ \|A(S) - \mathbf{w}^*\|_2, \frac{1}{n} \right\} \left( \sqrt{\frac{d}{n}} + \frac{d}{n} \right), \quad (2)$$

which is the optimal result when we only consider the order of $n$. The uniform convergence results are related to the dimension $d$, which are unacceptable in high-dimensional learning problems. Note that (2) requires an additional smoothness-type assumption. As a comparison, when $f$ is $\gamma$-smooth and $A$ is a deterministic algorithm, our result in Theorem 2 becomes

$$\|\nabla F(A(S)) - \nabla F_S(A(S))\|_2$$
$$\lesssim \beta \log n \log(1/\delta) + \frac{\log(1/\delta)}{n} + \sqrt{\frac{\mathbb{E}_Z \left[ \|\nabla f(\mathbf{w}^*; Z)\|_2^2 \right] \log(1/\delta)}{n}} + \|A(S) - \mathbf{w}^*\| \sqrt{\frac{\log(1/\delta)}{n}}.$$

Above inequality also holds in nonconvex problems and implies that when the uniformly stable in gradients parameter $\beta$ is smaller than $1/\sqrt{n}$, our bound is tighter than (2) and is dimension independent.

### 3.2 Sharper Excess Risk Bounds

In this subsection, we proceed to the desired upper bound of excess risk. The result is obtained by applying PL condition to Theorem 2 as in the following Theorem.

**Theorem 3.** *Let assumptions in Theorem 2 hold. Suppose the function $f$ is $\gamma$-smooth and the population risk $F$ satisfies the PL condition with parameter $\mu$. $\mathbf{w}^*$ is the projection of $A(S)$ onto the solution set $\arg\min_{\mathbf{w}\in\mathcal{W}} F(\mathbf{w})$. Then for any $\delta \in (0,1)$, when $n \geq \frac{32\gamma^2 \log(6/\delta)}{\mu^2}$, with probability at least $1-\delta$, we have*

$$\mathbb{E}_A[F(A(S))] - F(\mathbf{w}^*)$$
$$\lesssim \frac{\mathbb{E}_A\left[\|\nabla F_S(A(S))\|_2^2\right]}{\mu} + \frac{\gamma F(\mathbf{w}^*)\log(1/\delta)}{\mu n} + \frac{M^2\log^2(1/\delta)}{\mu n^2} + \frac{\beta^2 \log^2 n \log^2(1/\delta)}{\mu}.$$

**Remark 5.** Notably, in Theorem 3, the PL condition is imposed to the population risk. This can be hold for many cases within classical learning theory. For instance, the classical linear regression setup: $f(w; (x,y)) = \|y - \langle w, x\rangle\|^2$, where $y = \langle w^*, x\rangle + v$, with $v \sim N(0,1)$ and $x \sim N(0, \sigma^2 I_{d\times d})$. In this case, we have $\mu = 2$.

Theorem 3 implies that excess risk can be bound by the optimization gradient error $\|\nabla F_S(A(S))\|_2$ and uniform stability in gradients $\beta$. Notably, the theorem is applied to any algorithm with uniform stability in gradient. Latter, we will analyze the stability for specific algorithm in the next Section. Note that the assumption $F(\mathbf{w}^*) < O(1/n)$ is common and can be found in Srebro et al. [43], Lei and Ying [26], Liu et al. [30], Zhang et al. [55], Zhang and Zhou [54] when analyzes sharper bounds. In our bound, when $F(\mathbf{w}^*) < O(1/n)$, and the stability $\beta = O(1/n)$, we can obtain the desired $O(1/n^2)$ upper bound for excess risk when the empirical risk is sufficiently minimized by algorithm $A$.

To further compare our result with the optimal results based on algorithmic stability, i.e., [20], they only impose the assumption of bounded loss function, which is relative weaker than ours (smoothness and PL condition). However, our result is supposed to be sharper than their $\tilde{O}(\beta + 1/n)$, since ours has the potential to be $\tilde{O}(1/n^2)$ but their results are at most $\tilde{O}(1/n)$ even if the algorithm is stable enough. More than that, the imposed smoothness and PL conditions in our theorem are also used in Klochkov and Zhivotovskiy [20] to further analysis the algorithmic stability for PGD, whereas their excess risk bound becomes $\tilde{O}(1/n)$. However, our sharper bound indicates that they do not fully leverage these assumptions. The improvement is mainly from the novel stability in gradient during our analysis. This is also why our work can fully utilize these assumptions.

In addition, our results provide a more granular analysis dependent on optimal parameters. On one hand, when the algorithm's stability $\beta = O(1/\sqrt{n})$ the upper bound, according to [20], can at most reach the order of $\tilde{O}(1/\sqrt{n})$ due to the algorithm's stability constraints. In contrast, our result shows that even under the assumption that $(F(\mathbf{w}^*) = O(1)$, treating $F(\mathbf{w}^*)$ as a constant, we can achieve an order of $\tilde{O}(1/n)$ under the same algorithmic stability. On the other hand, their result is insensitive to the stability parameter being smaller than $O(1/n)$ and their best rates can only up to $\tilde{O}(1/n)$. Our results can be up to $\tilde{O}(1/n^2)$ under some specific assumptions. We will discuss it in Section 4.

While Theorem 3 considers the algorithm's uniform stability in gradients rather than the classic uniform stability, calculating the uniform stability in gradients is not more challenging than the classic uniform stability. We discuss uniform stability in gradients for common algorithms such as ERM, PGD, and SGD in Section 4. Our results can be easily extended to other stable algorithms. Due to the smoothness property linking uniform stability in gradients with uniform argument stability, many works [1, 12, 17] that explore uniform argument stability can also utilize our method.

Finally, we notice that there are currently several studies based on Klochkov and Zhivotovskiy [20] addressing different settings, such as differentially private models [18] and pairwise learning [27]. These studies also utilize the smoothness assumption in their optimization analysis. However, since they rely on the method established in Klochkov and Zhivotovskiy [20] for the generalization aspect of their research, they can only achieve an $\tilde{O}(1/n)$ bound at most. In contrast, they can easily achieve better results without making additional assumptions using our method.

## 4 Application

In this section, we take ERM, PGD, and SGD as examples to provide a detailed discussion of how our algorithm can be applied to common methods, resulting in tighter upper bounds.

## 4.1 Empirical Risk Minimizer

Empirical risk minimizer is one of the classical approaches for solving stochastic optimization (also referred to as sample average approximation (SAA)) in machine learning community. The following lemma shows the uniform stability in gradient for ERM under the PL condition and $\gamma$-smoothness assumptions.

**Lemma 1** (Stability of ERM). *Suppose the objective function $f$ is $M$-Lipschitz and $\gamma$-smooth, the empirical risk $F_S$ and $F_{S^{(i)}}$ satisfy the PL condition with parameter $\mu$. Let $\hat{\mathbf{w}}^*(S^{(i)})$ be the ERM of $F_{S^{(i)}}(\mathbf{w})$ that denotes the empirical risk on the samples $S^{(i)} = \{z_1, ..., z_i', ..., z_n\}$ and $\hat{\mathbf{w}}^*(S)$ be the ERM of $F_S(\mathbf{w})$ on the samples $S = \{z_1, ..., z_i, ..., z_n\}$. For any $S^{(i)}$ and $S$, there holds the following uniform stability bound of ERM:*

$$\forall z \in \mathcal{Z}, \quad \left\| \nabla f(\hat{\mathbf{w}}^*(S^{(i)}); z) - \nabla f(\hat{\mathbf{w}}^*(S); z) \right\|_2^2 \leq \frac{16 M^2 \gamma^2}{n^2 \mu^2}.$$

Then, we present the application of our main sharper Theorem 2. In the PL condition and smooth case, we provide a up to $\tilde{O}\left(1/n^2\right)$ high probability excess risk guarantee valid for any algorithms depending on the optimal population error $F(\mathbf{w}^*)$.

**Theorem 4.** *Let assumptions in Theorem 3 and Lemma 1 hold. Suppose the function $f$ is nonnegative. Then for any $\delta \in (0, 1)$, when $n \geq \frac{32 \gamma^2 \log (6/\delta)}{\mu^2}$, with probability at least $1 - \delta$, we have*

$$F(\hat{\mathbf{w}}^*(S)) - F(\mathbf{w}^*) \lesssim \frac{\gamma F(\mathbf{w}^*) \log (1/\delta)}{\mu n} + \frac{M^2 \gamma^2 \log^2 n \log^2(1/\delta)}{\mu^3 n^2}.$$

**Remark 6.** Theorem 4 shows that when the objective function $f$ is $M$-Lipschitz, $\gamma$-smooth and nonnegative, at the same time, both the empirical risk $F_S$ and the population risk $F$ satisfies PL condition with parameter $\mu$, then the high probability risk bounds can be up to $\tilde{O}\left(1/n^2\right)$ for ERM. The most related work to ours is Zhang et al. [55]. They also obtain the $\tilde{O}\left(1/n^2\right)$-type bounds for ERM by uniform convergence of gradients approach under the same assumptions. However, they need the sample number $n = \Omega(\gamma^2 d/\mu^2)$, which is related to the dimension $d$. Our risk bounds are dimension independent and only require the sample number $n = \Omega(\gamma^2/\mu^2)$. Comparing with Klochkov and Zhivotovskiy [20], we add two assumptions, smoothness and $F(\mathbf{w}^*) < O(1/n)$, the later of which is a common assumption towards sharper risk bounds [43, 26, 30, 55, 54] called low-noise, but our bounds are also tighter, from $\tilde{O}(1/n)$ to $\tilde{O}\left(1/n^2\right)$.

Our results are asymptotically optimal, which aligns with existing theories. According to the classical asymptotic theory, under some local regularity assumptions, when $n \to \infty$, it is shown in the asymptotic statistics monographs [45] that

$$\sqrt{n}(\hat{\mathbf{w}}^*(S) - \mathbf{w}^*) \xrightarrow{\rho} \mathcal{N}(0, \mathbf{H}^{-1} \mathbf{Q} \mathbf{H}^{-1}), \tag{3}$$

where $\hat{\mathbf{w}}^*(S)$ denotes the ERM algorithm, $\mathbf{H} = \nabla^2 F(\mathbf{w}^*)$, $\mathbf{Q}$ is the covariance matrix of the loss gradient at $\mathbf{w}^*$ (also called Fisher's information matrix): $\mathbf{Q} = \mathbb{E}[\nabla f(\mathbf{w}^*; z) \nabla f(\mathbf{w}^*; z)^T]$ ($\mathbf{A}^T$ denotes the transpose of a matrix $\mathbf{A}$), and $\xrightarrow{\rho}$ means convergence in distribution. The second-order Taylor expansion of the population risk around $\mathbf{w}^*$ then allows to derive the same asymptotic law for the scaled excess risk $2n(F(\hat{\mathbf{w}}^*(S)) - F(\mathbf{w}^*))$. Under suitable conditions, this asymptotic rate is usually theoretically optimal [44]. For example, when $f(\mathbf{w}; z)$ is a negative log-likelihood, this asymptotic rate matches the Hajek-LeCam asymptotic minimax lower bound [16, 23]. We then analyze the result in Theorem 4. In the proof of Theorem 3, before we use the self-bounded smoothness property $\|\nabla f(\mathbf{w}^*; z)\|^2 \leq 4\gamma f(\mathbf{w}^*; z)$, we get the following result for Theorem 4

$$F(\hat{\mathbf{w}}^*(S)) - F(\mathbf{w}^*) \lesssim \frac{\mathbb{E}[\|\nabla f(\mathbf{w}^*; z)\|^2] \log(1/\delta)}{\mu n} + \frac{M^2 \gamma^2 \log^2(n) \log^2(1/\delta)}{\mu^3 n^2}.$$

Our result is the finite sample version of the asymptotic rate (3), which characterizes the critical sample size sufficient to enter this "asymptotic regime". This is because the excess risk error $F(\hat{\mathbf{w}}^*(S)) - F(\mathbf{w}^*)$ can be approximated by the quadratic form $(\hat{\mathbf{w}}^*(S) - \mathbf{w}^*)^T \mathbf{H} (\hat{\mathbf{w}}^*(S) - \mathbf{w}^*)$. $1/\mu$ is a natural proxy for the inverse Hessian $\mathbf{H}^{-1}$, and $\mathbb{E}[\|\nabla f(\mathbf{w}^*; z)\|^2]$ is a natural proxy for Fisher's information matrix $\mathbf{Q}$. Furthermore, when discussing sample complexity, Xu and Zeevi [50] constructed a simple linear model to demonstrate the constant-level optimality of the sample complexity lower bound $\Omega(d\beta^2/\mu^2)$ under such conditions. Our theorem further reveals, through the use of stability methods, that this complexity lower bound can be independent of the dimensionality $d$.

## 4.2 Projected Gradient Descent

Note that when the objective function $f$ is smooth and the empirical risk $F_S$ satisfies the PL condition, the optimization error can be ignored. However, Klochkov and Zhivotovskiy [20] does not use smoothness assumption for their generalization bounds, which only derive high probability excess risk bound of order $\tilde{O}(1/n)$ after $T = O(\log(n))$ steps. In this subsection, we provide sharper risk bound under the same iteration steps, which is because our generalization analysis also fully utilized the smooth assumptions. Here we introduce the procedure of the PGD algorithm.

Let $\mathbf{w}_1 \in \mathbb{R}^d$ be an initial point and $\{\eta_t\}_t$ be a sequence of positive step sizes. PGD updates parameters by

$$\mathbf{w}_{t+1} = \Pi_{\mathcal{W}} \left( \mathbf{w}_t - \eta_t \nabla F_S \left( \mathbf{w}_t \right) \right),$$

where $\nabla F_S(\mathbf{w}_t)$ denotes a subgradient of $F_S$ w.r.t. $\mathbf{w}_t$ and $\Pi_{\mathcal{W}}$ is the projection operator onto $\mathcal{W}$.

**Lemma 2** (Stability of Projected Gradient Descent [12, 17]). *Suppose the objective function $f$ is $M$-Lipschitz and $\gamma$-smooth, the empirical risk $F_S$ and $F_{S^{(i)}}$ satisfy the PL condition with parameter $\mu$. Let $\mathbf{w}'_t$ be the output of $F_{S^{(i)}}(\mathbf{w})$ on $t$-th iteration on the samples $S^{(i)} = \{z_1, ..., z'_i, ..., z_n\}$ in running PGD, and $\mathbf{w}_t$ be the output of $F_S(\mathbf{w})$ on $t$-th iteration on the samples $S = \{z_1, ..., z_i, ..., z_n\}$ in running PGD. Let the constant step size $\eta_t = \frac{1}{\gamma}$. For any $S^{(i)}$ and $S$, there holds the following uniform stability bound of PGD:*

$$\forall z \in \mathcal{Z}, \quad \left\| \nabla f(\mathbf{w}^i_t; z) - \nabla f(\mathbf{w}_t; z) \right\|^2_2 \leq \frac{4M^2 \gamma^2}{n^2 \mu^2}.$$

The derivations of Feldman and Vondrak [12] in Section 4.1.2 (See also [17] in Section 3.4) imply that if the objective function $f$ is $\gamma$-smooth and $M$-Lipschitz, and the empirical risk $F_S$ and $F_{S^{(i)}}$ satisfy the PL condition with parameter $\mu$, then PGD with the constant step size $\eta = \frac{1}{\gamma}$ is $\left( \frac{2M}{n\mu} \right)$-uniformly argument stable for any number of steps, which means that PGD is $\left( \frac{2M\gamma}{n\mu} \right)$-uniformly-stable in gradients regardless of iteration steps.

**Theorem 5.** *Let assumptions in Theorem 3 and Lemma 1 hold. Suppose the function $f$ is nonnegative and $\mathbf{w}^*$ belongs to the projected set $\mathcal{W}$. Let $\{\mathbf{w}_t\}_t$ be the sequence produced by PGD with $\eta_t = 1/\gamma$. Then for any $\delta \in (0,1)$, when $n \geq \frac{32\gamma^2 \log(6/\delta)}{\mu^2}$, with probability at least $1 - \delta$, we have*

$$F(\mathbf{w}_{T+1}) - F(\mathbf{w}^*) \lesssim \left( 1 - \frac{\mu}{\gamma} \right)^{2T} + \frac{\gamma F(\mathbf{w}^*) \log(1/\delta)}{\mu n} + \frac{M^2 \gamma^2 \log^2 n \log^2(1/\delta)}{\mu^3 n^2}.$$

*Furthermore, assume $F(\mathbf{w}^*) < O(\frac{1}{n})$ and let $T \asymp \log(n)$, we have*

$$F(\mathbf{w}_{T+1}) - F(\mathbf{w}^*) \lesssim \frac{M^2 \gamma^2 \log^2 n \log^2(1/\delta)}{\mu^3 n^2}.$$

**Remark 7.** Theorem 5 shows that under the same assumptions as Klochkov and Zhivotovskiy [20], our bound is $O\left( \frac{F(\mathbf{w}^*) \log(1/\delta)}{n} + \frac{\log^2(n) \log^2(1/\delta)}{n^2} \right)$. Comparing with their bound $O\left( \frac{\log(n) \log(1/\delta)}{n} \right)$, we are sharper because $F(\mathbf{w}^*)$ is the minimal population risk, which is a common assumption towards sharper risk bounds [43, 26, 30, 55, 54]. We use this assumption to demonstrate that under low-noise conditions, our bounds can achieve the tightest possible rate of $\tilde{O}(1/n^2)$. This is because we also fully utilized the properties of smooth in the process of analyzing generalization errors. Considering that smooth is a very common assumption in optimization communities, our work can better utilize existing optimization work and obtain better excess risk bounds under the same assumptions.

## 4.3 Stochastic Gradient Descent

Stochastic gradient descent optimization algorithm has been widely used in machine learning due to its simplicity in implementation, low memory requirement and low computational complexity per iteration, as well as good practical behavior. We provide the excess risk bounds for SGD using our method in this subsection. Here we introduce the procedure of the standard SGD algorithm.

Let $\mathbf{w}_1 \in \mathbb{R}^d$ be an initial point and $\{\eta_t\}_t$ be a sequence of positive step sizes. SGD updates parameters by

$$\mathbf{w}_{t+1} = \Pi_{\mathcal{W}} \left( \mathbf{w}_t - \eta_t \nabla f \left( \mathbf{w}_t; z_{i_t} \right) \right),$$

where $\nabla f(\mathbf{w}_t; z_{i_t})$ denotes a subgradient of $f$ w.r.t. $\mathbf{w}_t$ and $i_t$ is independently drawn from the uniform distribution over $[n] := \{1, 2, \ldots, n\}$.

**Lemma 3** (Stability of SGD). *Suppose the objective function $f$ is $M$-Lipschitz and $\gamma$-smooth, the empirical risk $F_S$ and $F_{S^{(i)}}$ satisfy the PL condition with parameter $\mu$. Let $\mathbf{w}_t^i$ be the output of $F_{S^{(i)}}(\mathbf{w})$ on $t$-th iteration on the samples $S^{(i)} = \{z_1, ..., z_i', ..., z_n\}$ in running SGD with $\eta_t = \frac{2t+1}{2\mu(t+1)^2}$ and $\mathbf{w}_t$ be the output of $F_S(\mathbf{w})$ on $t$-th iteration on the samples $S = \{z_1, ..., z_i, ..., z_n\}$ in running SGD with $\eta_t = \frac{2t+1}{2\mu(t+1)^2}$. For any $S^{(i)}$ and $S$ and any $z \in \mathcal{Z}$, there holds the following uniform stability bound of SGD:*

$$\mathbb{E}_A\left[\left\|\nabla f(\mathbf{w}_t; z) - \nabla f(\mathbf{w}_t^i; z)\right\|_2^2\right] \leq \frac{6M^2\gamma^3}{t\mu^3} + \frac{48M^2\gamma^2}{n^2\mu^2}.$$

Next, we introduce a necessary assumption in stochastic optimization theory.

**Assumption 1.** *Assume the existence of $\sigma > 0$ satisfying*

$$\mathbb{E}_{i_t}[\|\nabla f(\mathbf{w}_t; z_{i_t}) - \nabla F_S(\mathbf{w}_t)\|_2^2] \leq \sigma^2, \quad \forall t \in \mathbb{N}, \tag{4}$$

*where $\mathbb{E}_{i_t}$ denotes the expectation w.r.t. $i_t$.*

**Remark 8.** Assumption 1 is a standard assumption from the stochastic optimization theory [34, 14, 15, 21, 56, 2, 25], which essentially bounds the variance of the stochastic gradients for dataset $S$.

**Theorem 6.** *Let Assumptions in Theorem 2 and Lemma 3 hold. Suppose Assumption 1 holds and the function $f$ is nonnegative. Let $\{\mathbf{w}_t\}_t$ be the sequence produced by SGD with $\eta_t = \frac{2t+1}{2\mu(t+1)^2}$. Then for any $\delta > 0$, when $n \geq \frac{32\gamma^2 \log(6/\delta)}{\mu^2}$, with probability at least $1 - \delta$, we have*

$$\mathbb{E}_A\left[F(\mathbf{w}_{T+1})\right] - F(\mathbf{w}^*) \lesssim \frac{\gamma F(\mathbf{w}^*) \log(1/\delta)}{\mu n} + \left(\frac{\gamma}{T\mu} + \frac{1}{n^2}\right) \frac{M^2\gamma^2 \log^2 n \log^2(1/\delta)}{\mu^3}.$$

*Furthermore, assume $T \asymp n^2$ and $F(\mathbf{w}^*) < O(\frac{1}{n})$, we have*

$$F(\mathbf{w}_{T+1}) - F(\mathbf{w}^*) \lesssim \frac{M^2\gamma^3 \log^2 n \log^2(1/\delta)}{\mu^4 n^2}.$$

**Remark 9.** Theorem 6 implies that high probability risk bounds for SGD optimization algorithm can be up to $\tilde{O}(1/n^2)$ and the rate is dimension-free in high-dimensional learning problems. We compare Theorem 6 with most related work. For algorithmic stability, high probability risk bounds in Fan and Lei [10] is up to $\tilde{O}(1/n)$ when choosing optimal iterate number $T$ for SGD optimization algorithm. To the best of knowledge, we are faster than all the existing bounds. The best high probability risk bounds $\tilde{O}(1/n^2)$ are given by [29] via uniform convergence, which require the same iterate number $T \asymp n^2$ and the sample number $n = \Omega(\gamma^2 d/\mu^2)$ depending on dimension $d$.

Although a considerable amount of theoretical (eg: [27]) and practical (eg: [35]) evidence indicates that multi-pass SGD can enhance the model's generalization performance. Someone may only utilize one-pass SGD in practice due to the large volume of data, which means that $T \asymp n$. In fact, when $T \asymp n$ and $F(\mathbf{w}^*) = O(1)$, our bound is $\tilde{O}(1/n)$, which is also the sharpest hight probability bound under $T \asymp n$ iterations. For comparison, there are two results in stability analysis that are similar to ours. One is the $\tilde{O}(1/n)$ result when $T \asymp n$ [26], but it pertains to the expected version and also needs $F(\mathbf{w}^*) = 0$. The high-probability version is significantly more challenging. Currently, the best result under the high-probability version is also $\tilde{O}(1/n)$ [29], but [29] requires $T \asymp n^2$ iterations.

## 5 Conclusion

In this paper, we derive sharper generalization bounds in gradients in nonconvex problems, which can further be used to obtain sharper high probability excess risk bounds for stable optimization algorithms. In application, we study three common algorithms: ERM, PGD, SGD. To the best of our knowledge, we provide the sharpest high probability dimension independent $\tilde{O}(1/n^2)$-type for these algorithms.

## Acknowledgments and Disclosure of Funding

This research was supported by National Key Research and Development Program of China (NO. 2024YFE0203200), National Natural Science Foundation of China (No.62476277), CCF-ALIMAMA TECH Kangaroo Fund(No.CCF-ALIMAMA OF 2024008), and Huawei-Renmin University joint program on Information Retrieval. We also acknowledge the support provided by the fund for building worldclass universities (disciplines) of Renmin University of China and by the funds from Beijing Key Laboratory of Big Data Management and Analysis Methods, Gaoling School of Artificial Intelligence, Renmin University of China, from Engineering Research Center of Next-Generation Intelligent Search and Recommendation, Ministry of Education, from Intelligent Social Governance Interdisciplinary Platform, Major Innovation & Planning Interdisciplinary Platform for the "DoubleFirst Class" Initiative, Renmin University of China, from Public Policy and Decision-making Research Lab of Renmin University of China, and from Public Computing Cloud, Renmin University of China.

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

## A  Additional definitions and lemmata

**Lemma 4** (Equivalence of tails and moments for random vectors [1]). *Let $X$ be a random variable with*

$$\|X\|_p \leq \sqrt{p}a + pb$$

*for some $a, b \geq 0$ and for any $p \geq 2$. Then for any $\delta \in (0,1)$ we have, with probability at least $1 - \delta$,*

$$|X| \leq e\left(a\sqrt{\log\left(\frac{e}{\delta}\right)} + b\log\frac{e}{\delta}\right).$$

**Lemma 5** (Vector Bernstein's inequality [36, 41]). *Let $\{X_i\}_{i=1}^n$ be a sequence of i.i.d. random variables taking values in a real separable Hilbert space. Assume that $\mathbb{E}[X_i] = \mu$, $\mathbb{E}[\|X_i - \mu\|^2] = \sigma^2$, and $\|X_i\| \leq M$, $\forall 1 \leq i \leq n$, then for all $\delta \in (0,1)$, with probability at least $1 - \delta$ we have*

$$\left\|\frac{1}{n}\sum_{i=1}^n X_i - \mu\right\| \leq \sqrt{\frac{2\sigma^2\log(\frac{2}{\delta})}{n}} + \frac{M\log\frac{2}{\delta}}{n}.$$

**Definition 4** (Weakly self-Bounded Function). *Assume that $a, b > 0$. A function $f : \mathcal{Z}^n \mapsto [0, +\infty)$ is said to be $(a, b)$-weakly self-bounded if there exist functions $f_i : \mathcal{Z}^{n-1} \mapsto [0, +\infty)$ that satisfies for all $Z^n \in \mathcal{Z}^n$,*

$$\sum_{i=1}^n (f_i(Z^n) - f(Z^n))^2 \leq af(Z^n) + b.$$

**Lemma 6** ([20]). *Suppose that $z_1, \ldots, z_n$ are independent random variables and the function $f : \mathcal{Z}^n \mapsto [0, +\infty)$ is $(a, b)$-weakly self-bounded and the corresponding function $f_i$ satisfy $f_i(Z^n) \geq f(Z^n)$ for $\forall i \in [n]$ and any $Z^n \in \mathcal{Z}^n$. Then, for any $t > 0$,*

$$Pr(\mathbb{E}f(z_1, \ldots, z_n) \geq f(z_1, \ldots, z_n) + t) \leq \exp\left(-\frac{t^2}{2a\mathbb{E}f(z_1, \ldots, z_n) + 2b}\right).$$

**Definition 5.** *A Rademacher random variable is a Bernoulli variable that takes values $\pm 1$ with probability $\frac{1}{2}$ each.*

## B  Summary of Our High Probability Excess Risk Bounds.

Our high probability excess risk bounds can be summarized in Table 1.

## C  Proofs of Section 3

### C.1  An Improved Moment Bound for Sums of Vector-valued Functions

In this section, we present our sharper moment bound for sums of vector-valued functions of $n$ independent variables, which is constant-level improvement comparing with [10]. With this inequality, we further prove the connection between algorithmic stability w.r.t. gradient and generalization. Notably, our bound relies on the "bounded difference" property, which is similar to uniform stability in gradient. Thus the bound can be further applied in deriving generalization bound on gradient, under stability condition.

**Lemma 7.** *Let $\mathbf{Z} = (Z_1, \ldots, Z_n)$ be a vector of independent random variables each taking values in $\mathcal{Z}$, $A$ is a independent random variable taking values in $\mathcal{A}$ and let $\mathbf{g}_1, \ldots, \mathbf{g}_n$ be some functions: $\mathbf{g}_i : \mathcal{Z}^n \times \mathcal{A} \mapsto \mathcal{H}$ such that the following holds for any $i \in [n]$:*

- $\|\mathbb{E}[\mathbf{g}_i(\mathbf{Z})|Z_i]\| \leq G$ *a.s.*

- $\mathbb{E}\left[\mathbf{g}_i(\mathbf{Z})|Z_{[n]\setminus\{i\}}\right] = 0$ *a.s.,*

- $\mathbf{g}_i$ *satisfies the bounded difference property with $\beta$, namely, for any $i = 1, \ldots, n$, the following inequality holds*

$$\sup_{z_1,\ldots,z_n,z_j'} \mathbb{E}_A\left[\|\mathbf{g}_i(z_1, \ldots, z_{j-1}, z_j, z_{j+1}, \ldots, z_n)\right.$$

$$\left. -\mathbf{g}_i(z_1, \ldots, z_{j-1}, z_j', z_{j+1}, \ldots, z_n)\|\right] \leq \beta. \tag{5}$$

*Then, for any $p \geq 2$, we have*

$$\left\| \left\| \sum_{i=1}^{n} \mathbf{g}_i \right\| \right\|_p \leq 4 \times 2^{\frac{1}{2p}} \left( \sqrt{\frac{p}{e}} \right) (\sqrt{2p} + 1) n \beta \lceil \log_2 n \rceil$$
$$+ 2(\sqrt{2p} + 1)\sqrt{n}G.$$

**Remark 10.** We compare with existing results. The proof is motivated by [4, 20]. [52, 53] have also explored several related problems based on this approach. However, all of them focus specifically on upper bounds for sums of real-valued functions. The result most closely related to Lemma 7 is provided by [10]. Under the same assumptions, [10] established the following inequality[2]

$$\left\| \left\| \sum_{i=1}^{n} \mathbf{g}_i \right\| \right\|_p \leq 4(\sqrt{2} + 1) n p \beta \lceil \log_2 n \rceil + 2(\sqrt{2} + 1)\sqrt{np}G. \tag{6}$$

It is easy to verify that our result is tighter than result provided by [10] for both the first and second term. Comparing Lemma 7 with (6), the larger $p$ is, the tighter our result is relative to (6). In the worst case, when $p = 2$, the constant of our second term is $0.879$ times tighter than (6), and the constant of our first term is $0.634$ times tighter than (6). This is because we derive the optimal Marcinkiewicz-Zygmund's inequality for random variables taking values in a Hilbert space in the proof.

Although Lemma 7 seems to the constant-level improvement, considering that this theorem has other broad applications, we report this result as well. On the other hand, the proof was challenging, as it involved establishing the best constant in Marcinkiewicz-Zygmund's inequality for random variables taking values in a Hilbert space, which has its foundations in Khintchine-Kahane's inequality. To prove the best constant, we utilized Stirling's formula for the Gamma function to construct appropriate functions, establishing both upper and lower bounds. Then using Mean Value Theorem, this approach ultimately led to the convergence of the constant as $p$ approaches infinity.

The proof of Lemma 7 is motivated by [4], which need the Marcinkiewicz-Zygmund's inequality for random variables taking values in a Hilbert space and the McDiarmid's inequality for vector-valued functions.

Firstly, we derive the optimal constants in the Marcinkiewicz-Zygmund's inequality for random variables taking values in a Hilbert space.

**Lemma 8** (Marcinkiewicz-Zygmund's Inequality for Random Variables Taking Values in a Hilbert Space). *Let $\mathbf{X}_1, \ldots, \mathbf{X}_n$ be random variables taking values in a Hilbert space with $\mathbb{E}[\mathbf{X}_i] = 0$ for all $i \in [n]$. Then for $p \geq 2$ we have*

$$\left\| \left\| \sum_{i=1}^{n} \mathbf{X}_i \right\| \right\|_p \leq 2 \cdot 2^{\frac{1}{2p}} \sqrt{\frac{np}{e}} \left( \frac{1}{n} \sum_{i=1}^{n} \left\| \|\mathbf{X}_i\| \right\|_p^p \right)^{\frac{1}{p}}.$$

**Remark 11.** Comparing with Marcinkiewicz-Zygmund's inequality given by [10], we provide best constants. Next, we give the proof of Lemma 8.

The Marcinkiewicz-Zygmund's inequality can be proved by using its connection to Khintchine-Kahane's inequality. Thus, we introduce the best constants in Khintchine-Kahane's inequality for random variables taking values from a Hilbert space here.

**Lemma 9** (Best constants in Khintchine-Kahane's inequality in Hilbert space [22, 33]). *For all $p \in [2, \infty)$ and for all choices of Hilbert space $\mathcal{H}$, finite sets of vectors $\mathbf{X}_i, \ldots, \mathbf{X}_n \in \mathcal{X} \in \mathcal{H}$, and independent Rademacher variables $r_1, \ldots, r_n$,*

$$\left[ \mathbb{E} \left\| \sum_{i=1}^{n} r_i \mathbf{X}_i \right\|^p \right]^{\frac{1}{p}} \leq C_p \cdot \left[ \sum_{i=1}^{n} \|\mathbf{X}_i\|^2 \right]^{\frac{1}{2}},$$

*where $C_p = 2^{\frac{1}{2}} \left\{ \frac{\Gamma\left(\frac{p+1}{2}\right)}{\sqrt{\pi}} \right\}^{\frac{1}{p}}$.*

---

[2]They assume $n = 2^k$, $k \in \mathbb{N}$. Here we give the version of their result with general $n$.

*Proof of Lemma 8.* The symmetrization argument goes as follows: Let $(r_1, \ldots, r_n)$ be i.i.d. with $\mathbb{P}(r_i = 1) = \mathbb{P}(r_i = -1) = 1/2$ and besides such that $r_1, \ldots, r_n$ and $(\mathbf{X}_1, \ldots, \mathbf{X}_n)$ are independent. Then by independence and symmetry, according to Lemma 1.2.6 of [7], conditioning on $(\mathbf{X}_1, \ldots, \mathbf{X}_n)$ yields

$$\mathbb{E}\left[\left\|\sum_{i=1}^{n} \mathbf{X}_i\right\|^p\right] = 2^p \mathbb{E}\left[\left\|\sum_{i=1}^{n} r_i \mathbf{X}_i\right\|^p\right] \leq 2^p \mathbb{E}\left[\mathbb{E}\left[\left\|\sum_{i=1}^{n} r_i \mathbf{X}_i\right\|^p \,\middle|\, \mathbf{X}_1, \ldots, \mathbf{X}_n\right]\right]. \tag{7}$$

As for the conditional expectation in (7), notice that by independence

$$\mathbb{E}\left[\left\|\sum_{i=1}^{n} r_i \mathbf{X}_i\right\|^p \,\middle|\, \mathbf{X}_1 = \mathbf{x}_1, \ldots, \mathbf{X}_n = \mathbf{x}_n\right] = \mathbb{E}\left[\left\|\sum_{i=1}^{n} r_i \mathbf{x}_i\right\|^p\right] \tag{8}$$

According to Lemma 9, for $v_n$-almost every $\mathbf{x}_1, \ldots, \mathbf{x}_n \in \mathcal{X}^n$, where $v_n := \mathbb{P} \circ (\mathbf{X}_1, \ldots, \mathbf{X}_n)^{-1}$ denotes the distribution of $(\mathbf{X}_1, \ldots, \mathbf{X}_n)$, we have

$$\left[\mathbb{E}\left\|\sum_{i=1}^{n} r_i \mathbf{x}_i\right\|^p\right] \leq C \cdot \left[\sum_{i=1}^{n} \|\mathbf{x}_i\|^2\right]^{\frac{p}{2}}, \tag{9}$$

where $C = 2^{\frac{p}{2}} \frac{\Gamma\left(\frac{p+1}{2}\right)}{\sqrt{\pi}}$ and $C$ is optimal. This means that for any constant $C'$ such that

$$\left[\mathbb{E}\left\|\sum_{i=1}^{n} r_i \mathbf{x}_i\right\|^p\right] \leq C' \cdot \left[\sum_{i=1}^{n} \|\mathbf{x}_i\|^2\right]^{\frac{p}{2}}, \tag{10}$$

for all $n \in \mathbb{N}$ and for each collection of vectors $\mathbf{x}_1, \ldots, \mathbf{x}_n$, it follows that $C' \geq C$.

From (8) and (9), we can infer that

$$\mathbb{E}\left[\left\|\sum_{i=1}^{n} r_i \mathbf{X}_i\right\|^p \,\middle|\, \mathbf{X}_1 = \mathbf{x}_1, \ldots, \mathbf{X}_n = \mathbf{x}_n\right] \leq C \cdot \left[\sum_{i=1}^{n} \|\mathbf{X}_i\|^2\right]^{\frac{p}{2}}.$$

Taking expectations in the above inequalities and (7) yield that

$$\mathbb{E}\left[\left\|\sum_{i=1}^{n} \mathbf{X}_i\right\|^p\right] \leq C \cdot \mathbb{E}\left[\sum_{i=1}^{n} \|\mathbf{X}_i\|^2\right]^{\frac{p}{2}}. \tag{11}$$

To see optimality let the above statement hold for some constants $C'$ in place of $C$. Then if we choose $\mathbf{X}_i := \mathbf{x}_i r_i, 1 \leq i \leq n$ with arbitrary reals vectors $\mathbf{x}_1, \ldots, \mathbf{x}_n$, it follows that

$$\mathbb{E}\left[\left\|\sum_{i=1}^{n} r_i \mathbf{x}_i\right\|^p\right] \leq C' \cdot \mathbb{E}\left[\sum_{i=1}^{n} \|\mathbf{x}_i\|^2\right]^{\frac{p}{2}},$$

whence we can conclude from (10) that $C' \geq C$. Thus we obtain that $C' = C$.

Notice that by Holder's inequality

$$\left[\sum_{i=1}^{n} \|\mathbf{X}_i\|^2\right]^{\frac{p}{2}} \leq n^{p/2-1} \sum_{i=1}^{n} \|\mathbf{X}_i\|^p. \tag{12}$$

Plugging (12) into (11), we have

$$\mathbb{E}\left[\left\|\sum_{i=1}^{n} \mathbf{X}_i\right\|^p\right] \leq C \cdot 2^p n^{p/2-1} \cdot \mathbb{E}\left[\sum_{i=1}^{n} \|\mathbf{X}_i\|^p\right],$$

where $C = 2^{\frac{p}{2}} \frac{\Gamma\left(\frac{p+1}{2}\right)}{\sqrt{\pi}}$ is a constant.

Next, we use the following form of Stirling's formula for the Gamma-function, which follows from (6.1.5), (6.1.15) and (6.1.38) in [6] to bound the constant $C$. For every $x > 0$, there exists a $\mu(x) \in (0, 1/(12x))$ such that

$$\Gamma(x) = \sqrt{2\pi} x^{x-1/2} e^{-x} e^{\mu(x)}.$$

Thus

$$C = 2^{\frac{p}{2}} \frac{\Gamma\left(\frac{p+1}{2}\right)}{\sqrt{\pi}} = g(p)\sqrt{2} e^{-p/2} p^{p/2},$$

with $g(p) = \left(1 + \frac{1}{p}\right)^{p/2} e^{v(p)-1/2}$, where $0 < v(p) < 1/(6(p+1))$. By Taylor's formula we have that

$$\log(1+x) = \sum_{m=1}^{\infty} \frac{1}{m} (-1)^{m-1} x^m, \quad \forall x \in (-1, 1],$$

and that for every $k \in \mathbb{N}_0$

$$\sum_{m=1}^{2k} \frac{1}{m}(-1)^{m-1} x^m \le \log(1+x) \le \sum_{m=1}^{2k+1} \frac{1}{m}(-1)^{m-1} x^m, \forall x \ge 0.$$

Therefor we obtain with $k = 1$ that

$$\log g(p) = \frac{p}{2}\log(1 + \frac{1}{p}) + v(p) - \frac{1}{2} \le -\frac{1}{4p} + \frac{1}{6p^2} + \frac{1}{6(p+1)} \le -\frac{1}{18p},$$

where the last equality follows from elementary calculus. Similarly,

$$\log g(p) = \frac{p}{2}\log(1 + \frac{1}{p}) + v(p) - \frac{1}{2} \ge -\frac{1}{4p} + v(p) \ge -\frac{1}{4p},$$

Thus, we have

$$e^{-\frac{1}{4p}}\sqrt{2}e^{-p/2}p^{p/2} < C < e^{-\frac{1}{18p}}\sqrt{2}e^{-p/2}p^{p/2},$$

which implies that $C$ is strictly smaller than $\sqrt{2}e^{-p/2}p^{p/2}$ for all $p \ge 2$.

Since $C = \frac{1}{g(p)}\sqrt{2}e^{-p/2}p^{p/2}$ and $g(p) \ge e^{-\frac{1}{4p}}$ , we can obtain that the relative error between $C$ and $\sqrt{2}e^{-p/2}p^{p/2}$ is equal to

$$\frac{1}{g(p)} - 1 \le e^{-\frac{1}{4p}} - 1 \le \frac{1}{4p} e^{\frac{1}{4p}}$$

using Mean Value Theorem. This implies that the corresponding relative errors between $C$ and $\sqrt{2}e^{-p/2}p^{p/2}$ converge to zero as $p$ tends to infinity.

The proof is complete.

$\square$

Then we introduce the McDiarmid's inequality for vector-valued functions. We firstly consider real-valued functions, which follows from the standard tail-bound of McDiarmid's inequality and Proposition 2.5.2 in [49].

**Lemma 10** (McDiarmid's Inequality for real-valued functions). *Let $Z_i, \ldots, Z_n$ be independent random variables, and $f : \mathcal{Z}^n \mapsto \mathbb{R}$ such that the following inequality holds for any $z_i, \ldots, z_{i-1}, z_{i+1}, \ldots, z_n$*

$$\sup_{z_i, z_i'} |f(z_1, \ldots, z_{i-1}, z_i, z_{i+1}, \ldots, z_n) - f(z_1, \ldots, z_{i-1}, z_i', z_{i+1}, \ldots, z_n)| \le \beta,$$

*Then for any $p > 1$ we have*

$$\|f(Z_1, \ldots, Z_n) - \mathbb{E}f(Z_1, \ldots, Z_n)\|_p \le \sqrt{2pn}\beta.$$

To derive the McDiarmid's inequality for vector-valued functions, we need the expected distance between $\mathbf{f}(Z_1, \ldots, Z_n)$ and its expectation.

**Lemma 11** ([38]). *Let $Z_i, \ldots, Z_n$ be independent random variables, and $\mathbf{f} : \mathcal{Z}^n \mapsto \mathcal{H}$ is a function into a Hilbert space $\mathcal{H}$ such that the following inequality holds for any $z_i, \ldots, z_{i-1}, z_{i+1}, \ldots, z_n$*

$$\sup_{z_i, z_i'} \|\mathbf{f}(z_1, \ldots, z_{i-1}, z_i, z_{i+1}, \ldots, z_n) - \mathbf{f}(z_1, \ldots, z_{i-1}, z_i', z_{i+1}, \ldots, z_n)\| \leq \beta,$$

*Then we have*

$$\mathbb{E}\left[\|\mathbf{f}(Z_1, \ldots, Z_n) - \mathbb{E}\mathbf{f}(Z_1, \ldots, Z_n)\|\right] \leq \sqrt{n}\beta.$$

Now, we can easily derive the $p$-norm McDiarmid's inequality for vector-valued functions which refines from [10] with better constants.

**Lemma 12** (McDiarmid's inequality for vector-valued functions). *Let $Z_i, \ldots, Z_n$ be independent random variables, and $\mathbf{f} : \mathcal{Z}^n \times \mathcal{A} \mapsto \mathcal{H}$ is a function into a Hilbert space $\mathcal{H}$ such that the following inequality holds for any $z_i, \ldots, z_{i-1}, z_{i+1}, \ldots, z_n$*

$$\sup_{z_i, z_i'} \mathbb{E}_A \|\mathbf{f}(z_1, \ldots, z_{i-1}, z_i, z_{i+1}, \ldots, z_n) - \mathbf{f}(z_1, \ldots, z_{i-1}, z_i', z_{i+1}, \ldots, z_n)\| \leq \beta, \quad (13)$$

*Then for any $p > 1$ we have*

$$\|\|\mathbb{E}_A\mathbf{f}(Z_1, \ldots, Z_n) - \mathbb{E}\mathbf{f}(Z_1, \ldots, Z_n)\|\|_p \leq (\sqrt{2p} + 1)\sqrt{n}\beta.$$

*Proof of Lemma 12.* Define a real-valued function $h : \mathcal{Z}^n \mapsto \mathbb{R}$ as

$$h(z_1, \ldots, z_n) = \|\mathbb{E}_A\mathbf{f}(z_1, \ldots, z_n) - \mathbb{E}[\mathbf{f}(Z_1, \ldots, Z_n)]\|.$$

We notice that this function satisfies the increment condition. For any $i$ and $z_1, \ldots, z_{i-1}, z_{i+1}, \ldots, z_n$, we have

$$\sup_{z_i, z_i'} |h(z_1, \ldots, z_{i-1}, z_i, z_{i+1}, \ldots, z_n) - h(z_1, \ldots, z_{i-1}, z_i', z_{i+1}, \ldots, z_n)|$$

$$= \sup_{z_i, z_i'} |\|\mathbb{E}_A\mathbf{f}(z_1, \ldots, z_n) - \mathbb{E}[\mathbf{f}(Z_1, \ldots, Z_n)]\| - \|\mathbb{E}_A\mathbf{f}(z_1, \ldots, z_{i-1}, z_i', z_{i+1}, \ldots, z_n) - \mathbb{E}[\mathbf{f}(Z_1, \ldots, Z_n)]\||$$

$$\leq \sup_{z_i, z_i'} \mathbb{E}_A\|\|\mathbf{f}(z_1, \ldots, z_n) - \mathbf{f}(z_1, \ldots, z_{i-1}, z_i', z_{i+1}, \ldots, z_n)\| \leq \beta.$$

Therefore, we can apply Lemma 10 to the real-valued function $h$ and derive the following inequality

$$\|h(Z_1, \ldots, Z_n) - \mathbb{E}[h(Z_1, \ldots, Z_n)]\|_p \leq \sqrt{2pn}\beta.$$

According to Lemma 11, we know the following inequality $\mathbb{E}[h(Z_1, \ldots, Z_n)] \leq \sqrt{n}\beta$. Combing the above two inequalities together and we can derive the following inequality

$$\|\|\mathbb{E}_A\mathbf{f}(Z_1, \ldots, Z_n) - \mathbb{E}\mathbf{f}(Z_1, \ldots, Z_n)\|\|_p$$
$$\leq \|h(Z_1, \ldots, Z_n) - \mathbb{E}[h(Z_1, \ldots, Z_n)]\|_p + \|\mathbb{E}[h(Z_1, \ldots, Z_n)]\|_p$$
$$\leq (\sqrt{2p} + 1)\sqrt{n}\beta.$$

The proof is complete.

$\square$

*Proof of Lemma 7.* For $\mathbf{g}(Z_1, \ldots, Z_n)$ and $D \subset [n]$, we write $\|\|\mathbf{g}\|\|_p(Z_D) = (\mathbb{E}[\|f\|^p \, Z_D])^{\frac{1}{p}}$. Without loss of generality, we suppose that $n = 2^k$. Otherwise, we can add extra functions equal to zero, increasing the number of therms by at most two times.

Consider a sequence of partitions $\mathcal{P}_0, \ldots, \mathcal{P}_k$ with $\mathcal{P}_0 = \{\{i\} : i \in [n]\}, \mathcal{P}_k$ with $\mathcal{P}_n = \{[n]\}$, and to get $\mathcal{P}_l$ from $\mathcal{P}_{l+1}$ we split each subset in $\mathcal{P}_{l+1}$ into two equal parts. We have

$$\mathcal{P}_0 = \{\{1\}, \ldots, \{2^k\}\}, \quad \mathcal{P}_1 = \{\{1, 2\}, \{3, 4\}, \ldots, \{2^k - 1, 2^k\}\}, \quad \mathcal{P}_k = \{\{1, \ldots, 2^k\}\}.$$

We have $|\mathcal{P}_l| = 2^{k-l}$ and $|P| = 2^l$ for each $P \in \mathcal{P}_l$. For each $i \in [n]$ and $l = 0, \dots, k$, denote by $P^l(i) \in \mathcal{P}_l$ the only set from $\mathcal{P}_l$ that contains $i$. In particular, $P^0(i) = \{i\}$ and $P^K(i) = [n]$.

For each $i \in [n]$ and every $l = 0, \dots, k$ consider the random variables

$$\mathbf{g}_i^l = \mathbf{g}_i^l(Z_i, Z_{[n]\setminus P^l(i)}) = \mathbb{E}[\mathbf{g}_i | Z_i, Z_{[n]\setminus P^l(i)}],$$

i.e. conditioned on $z_i$ and all the variables that are not in the same set as $Z_i$ in the partition $\mathcal{P}_l$. In particular, $\mathbf{g}_i^0 = \mathbf{g}_i$ and $\mathbf{g}_i^k = \mathbb{E}[\mathbf{g}_i | Z_i]$. We can write a telescopic sum for each $i \in [n]$,

$$\mathbf{g}_i - \mathbb{E}[\mathbf{g}_i | Z_i] = \sum_{l=1}^{k-1} \mathbf{g}_i^l - \mathbf{g}_i^{l+1}.$$

Then, by the triangle inequality

$$\left\|\left\|\left\| \sum_{i=1}^n \mathbf{g}_i \right\|\right\|\right\|_p \leq \left\|\left\|\left\| \sum_{i=1}^n \mathbb{E}[\mathbf{g}_i | Z_i] \right\|\right\|\right\|_p + \sum_{l=0}^{k-1} \left\|\left\|\left\| \sum_{i=1}^n \mathbf{g}_i^l - \mathbf{g}_i^{l+1} \right\|\right\|\right\|_p. \tag{14}$$

To bound the first term, since $\|\mathbb{E}[\mathbf{g}_i | Z_i]\| \leq G$, we can check that the vector-valued function $\mathbf{f}(Z_1, \dots, Z_n) = \sum_{i=1}^n \mathbb{E}[\mathbf{g}_i | Z_i]$ satisfies (13) with $\beta = 2G$, and $\mathbb{E}[\mathbb{E}[\mathbf{g}_i | Z_i]] = 0$, applying Lemma 12 with $\beta = 2G$, we have

$$\left\|\left\|\left\| \sum_{i=1}^n \mathbb{E}[\mathbf{g}_i | Z_i] \right\|\right\|\right\|_p \leq 2(\sqrt{2p} + 1)\sqrt{n}G. \tag{15}$$

Then we start to bound the second term of the right hand side of (14). Observe that

$$\mathbf{g}_i^{l+1}(Z_i, Z_{[n]\setminus P^{l+1}(i)}) = \mathbb{E}\left[ \mathbf{g}_i^l(Z_i, Z_{[n]\setminus P^l(i)}) \middle| Z_i, Z_{[n]\setminus P^{l+1}(i)} \right],$$

where the expectation is taken with respect to the variables $Z_j, j \in P^{l+1}(i)\setminus P^l(i)$. Changing any $Z_j$ would change $\mathbf{g}_i^l$ by $\beta$. Therefore, we apply Lemma 12 with $\mathbf{f} = \mathbf{g}_i^l$ where there are $2^l$ random variables and obtain a uniform bound

$$\left\|\left\|\left\| \mathbf{g}_i^l - \mathbf{g}_i^{l+1} \right\|\right\|\right\|_p (Z_i, Z_{[n]\setminus P^{l+1}(i)}) \leq (\sqrt{2p} + 1)\sqrt{2^l}\beta, \quad \forall p \geq 2,$$

Taking integration over $(Z_i, Z_{[n]\setminus P^{l+1}(i)})$, we have $\left\|\left\|\left\| \mathbf{g}_i^l - \mathbf{g}_i^{l+1} \right\|\right\|\right\|_p \leq (\sqrt{2p} + 1)\sqrt{2^l}\beta$ as well.

Next, we turn to the sum $\sum_{i \in P^l} \mathbf{g}_i^l - \mathbf{g}_i^{l+1}$ for any $P^l \in \mathcal{P}_l$. Since $\mathbf{g}_i^l - \mathbf{g}_i^{l+1}$ for $i \in P^l$ depends only on $Z_i, Z_{[n]\setminus P^l}$, the terms are independent and zero mean conditioned on $Z_{[n]\setminus P^l}$. Applying Lemma 8, we have for any $p \geq 2$,

$$\left\|\left\|\left\| \sum_{i \in P^l} \mathbf{g}_i^l - \mathbf{g}_i^{l+1} \right\|\right\|\right\|_p^p (Z_{[n]\setminus P^l}) \leq \left( 2 \cdot 2^{\frac{1}{2p}} \sqrt{\frac{2^l p}{e}} \right)^p \frac{1}{2^l} \sum_{i \in P^l} \left\|\left\|\left\| \mathbf{g}_i^l - \mathbf{g}_i^{l+1} \right\|\right\|\right\|_p^p (Z_{[n]\setminus P^l}).$$

Integrating with respect to $(Z_{[n]\setminus P^l})$ and using $\left\|\left\|\left\| \mathbf{g}_i^l - \mathbf{g}_i^{l+1} \right\|\right\|\right\|_p \leq (\sqrt{2p} + 1)\sqrt{2^l}\beta$, we have

$$\left\|\left\|\left\| \sum_{i \in P^l} \mathbf{g}_i^l - \mathbf{g}_i^{l+1} \right\|\right\|\right\|_p \leq \left( 2 \cdot 2^{\frac{1}{2p}} \sqrt{\frac{2^l p}{e}} \right) \frac{1}{2^l} \times 2^l (\sqrt{2p} + 1)\sqrt{2^l}\beta$$

$$= 2^{1+\frac{1}{2p}} \left( \sqrt{\frac{p}{e}} \right) (\sqrt{2p} + 1) 2^l \beta.$$

Then using triangle inequality over all sets $P^l \in \mathcal{P}_l$, we have

$$\left\|\left\|\left\| \sum_{i \in [n]} \mathbf{g}_i^l - \mathbf{g}_i^{l+1} \right\|\right\|\right\|_p \leq \sum_{P^l \in \mathcal{P}_l} \left\|\left\|\left\| \sum_{i \in P^l} \mathbf{g}_i^l - \mathbf{g}_i^{l+1} \right\|\right\|\right\|_p$$

$$\leq 2^{k-l} \times 2^{1+\frac{1}{2p}} \left( \sqrt{\frac{p}{e}} \right) (\sqrt{2p} + 1) 2^l \beta$$

$$\leq 2^{1+\frac{1}{2p}} \left( \sqrt{\frac{p}{e}} \right) (\sqrt{2p} + 1) 2^k \beta.$$

Recall that $2^k \leq n$ due to the possible extension of the sample. Then we have

$$\sum_{l=0}^{k-1} \left\| \left\| \sum_{i=1}^{n} \mathbf{g}_i^l - \mathbf{g}_i^{i+1} \right\| \right\|_p \leq 2^{2+\frac{1}{2p}} \left( \sqrt{\frac{p}{e}} \right) (\sqrt{2p} + 1) n \beta \lceil \log_2 n \rceil .$$

We can plug the above bound together with (15) into (14), to derive the following inequality

$$\left\| \left\| \sum_{i=1}^{n} \mathbf{g}_i \right\| \right\|_p \leq 2(\sqrt{2p} + 1)\sqrt{n}G + 2^{2+\frac{1}{2p}} \left( \sqrt{\frac{p}{e}} \right) (\sqrt{2p} + 1) n \beta \lceil \log_2 n \rceil .$$

The proof is completed.

$\square$

## C.2 Proofs of Subsection 3.1

*Proof of Theorem 1.* Let $S = \{z_1, \ldots, z_n\}$ be a set of independent random variables each taking values in $\mathcal{Z}$ and $S' = \{z'_1, \ldots, z'_n\}$ be its independent copy. For any $i \in [n]$, define $S^{(i)} = \{z_i, \ldots, z_{i-1}, z'_i, z_{i+1}, \ldots, z_n\}$ be a dataset replacing the $i$-th sample in $S$ with another i.i.d. sample $z'_i$. Then we can firstly write the following decomposition

$$n\nabla F(A(S)) - n\nabla F_S(A(S))$$

$$= \sum_{i=1}^{n} \mathbb{E}_Z \left[ \nabla f(A(S); Z)] - \mathbb{E}_{z'_i} \left[ \nabla f(A(S^{(i)}), Z) \right] \right]$$

$$+ \sum_{i=1}^{n} \mathbb{E}_{z'_i} \left[ \mathbb{E}_Z \left[ \nabla f(A(S^{(i)}), Z) \right] - \nabla f(A(S^{(i)}), z_i) \right]$$

$$+ \sum_{i=1}^{n} \mathbb{E}_{z'_i} \left[ \nabla f(A(S^{(i)}), z_i) \right] - \sum_{i=1}^{n} \nabla f(A(S), z_i).$$

We denote that $\mathbf{g}_i(S) = \mathbb{E}_{z'_i} \left[ \mathbb{E}_Z \left[ \nabla f(A(S^{(i)}), Z) \right] - \nabla f(A(S^{(i)}), z_i) \right]$, thus we have

$$\mathbb{E}_A \left[ \| n\nabla F(A(S)) - n\nabla F_S(A(S)) \|_2 \right]$$

$$= \mathbb{E}_A \left\| \sum_{i=1}^{n} \mathbb{E}_Z \left[ \nabla f(A(S); Z)] - \mathbb{E}_{z'_i} \left[ \nabla f(A(S^{(i)}), Z) \right] \right] \right.$$

$$+ \sum_{i=1}^{n} \mathbb{E}_{z'_i} \left[ \mathbb{E}_Z \left[ \nabla f(A(S^{(i)}), Z) \right] - \nabla f(A(S^{(i)}), z_i) \right] \qquad (16)$$

$$\left. + \sum_{i=1}^{n} \mathbb{E}_{z'_i} \left[ \nabla f(A(S^{(i)}), z_i) \right] - \sum_{i=1}^{n} \nabla f(A(S), z_i) \right\|_2$$

$$\leq 2n\beta + \mathbb{E}_A \left[ \left\| \sum_{i=1}^{n} \mathbf{g}_i(S) \right\|_2 \right],$$

where the inequality holds from the definition of uniform stability in gradients.

According to our assumptions, we get $\|\mathbf{g}_i(S)\|_2 \leq 2M$ and

$$\mathbb{E}_{z_i}[\mathbf{g}_i(S)] = \mathbb{E}_{z_i} \mathbb{E}_{z'_i} \left[ \mathbb{E}_Z \left[ \nabla f(A(S^{(i)}); Z) \right] - \nabla f(A(S^{(i)}); z_i) \right]$$

$$= \mathbb{E}_{z'_i} \left[ \mathbb{E}_Z \left[ \nabla f(A(S^{(i)}); Z) \right] - \mathbb{E}_{z_i} \left[ \nabla f(A(S^{(i)}); z_i) \right] \right] = 0,$$

where this equality holds from the fact that $z_i$ and $Z$ follow from the same distribution. For any $i \in [n]$, any $j \neq i$ and any $z_j''$, we have

$$\mathbb{E}_A \left[ \left\| \mathbf{g}_i(z_1, \ldots, z_{j-1}, z_j, z_{j+1}, \ldots, z_n) - \mathbf{g}_i(z_1, \ldots, z_{j-1}, z_j'', z_{j+1}, \ldots, z_n) \right\|_2 \right]$$

$$\leq \mathbb{E}_A \left[ \left\| \mathbb{E}_{z_i'} \left[ \mathbb{E}_Z \left[ \nabla f(A(S^{(i)}); Z) \right] - \nabla f(A(S^{(i)}); z_i) \right] - \mathbb{E}_{z_i'} \left[ \mathbb{E}_Z \left[ \nabla f(A(S_j^{(i)}); Z) \right] - \nabla f(A(S_j^{(i)}); z_i) \right] \right\|_2 \right]$$

$$\leq \mathbb{E}_A \left[ \left\| \mathbb{E}_{z_i'} \left[ \mathbb{E}_Z \left[ \nabla f(A(S^{(i)}); Z) - \nabla f(A(S_j^{(i)}); Z) \right] \right] \right\|_2 \right]$$

$$+ \mathbb{E}_A \left[ \left\| \mathbb{E}_{z_i'} \left[ \mathbb{E}_Z \left[ \nabla f(A(S^{(i)}); Z) \right] - \nabla f(A(S_j^{(i)}); z_i) \right] \right\|_2 \right]$$

$$\leq 2\beta,$$

where $S^{(i)} = \{z_i, \ldots, z_{i-1}, z_i', z_{i+1}, \ldots, z_n\}$. Since the noise introduced by the algorithm's inherent randomness $A$ is typically assumed to be independent of the dataset $S$, we have verified that three conditions in Lemma 7 are satisfied for $\mathbf{g}_i(S)$. We have the following result for any $p > 2$

$$\left\| \left\| \sum_{i=1}^n \mathbf{g}_i(S) \right\| \right\|_p \leq 4(\sqrt{2p} + 1)\sqrt{n}M + 8 \times 2^{\frac{1}{4}} \left( \sqrt{\frac{p}{e}} \right) (\sqrt{2p} + 1)n\beta \lceil \log_2 n \rceil .$$

According to Lemma 4 for any $\delta \in (0, 1)$, with probability at least $1 - \delta$, we have

$$\left\| \sum_{i=1}^n \mathbf{g}_i(S) \right\|_2$$

$$\leq 4\sqrt{n}M + 8 \times 2^{\frac{3}{4}} \sqrt{e}n\beta \lceil \log_2 n \rceil \log(e/\delta) + (4e\sqrt{2n}M + 8 \times 2^{\frac{1}{4}} \sqrt{e}n\beta \lceil \log_2 n \rceil) \sqrt{\log e/\delta}.$$

We can finally combine the above inequality and (16) to derive that, for any $\delta \in (0, 1)$, with probability at least $1 - \delta$, we have

$$\mathbb{E}_A \left[ \| \nabla F(A(S)) - \nabla F_S(A(S)) \|_2 \right]$$

$$\leq 2\beta + \frac{4M \left( 1 + e\sqrt{2\log(e/\delta)} \right)}{\sqrt{n}} + 8 \times 2^{\frac{1}{4}} (\sqrt{2} + 1)\sqrt{e}\beta \lceil \log_2 n \rceil \log(e/\delta).$$

The proof is completed. $\qquad\square$

*Proof of Theorem 2.* We can firstly write the following decomposition

$$n\nabla F(A(S)) - n\nabla F_S(A(S))$$

$$= \sum_{i=1}^n \mathbb{E}_Z \left[ \nabla f(A(S); Z) - \mathbb{E}_{z_i'} \left[ \nabla f(A(S^{(i)}), Z) \right] \right]$$

$$+ \sum_{i=1}^n \mathbb{E}_{z_i'} \left[ \mathbb{E}_Z \left[ \nabla f(A(S^{(i)}), Z) \right] - \nabla f(A(S^{(i)}), z_i) \right]$$

$$+ \sum_{i=1}^n \mathbb{E}_{z_i'} \left[ \nabla f(A(S^{(i)}), z_i) \right] - \sum_{i=1}^n \nabla f(A(S), z_i).$$

We denote that $\mathbf{h}_i(S) = \mathbb{E}_{z_i'} \left[ \mathbb{E}_Z \left[ \nabla f(A(S^{(i)}), Z) \right] - \nabla f(A(S^{(i)}), z_i) \right]$, we have

$$n\nabla F(A(S)) - n\nabla F_S(A(S)) - \sum_{i=1}^n \mathbf{h}_i(S)$$

$$= \sum_{i=1}^n \mathbb{E}_Z \left[ \nabla f(A(S); Z) - \mathbb{E}_{z_i'} \left[ \nabla f(A(S^{(i)}), Z) \right] \right]$$

$$+ \sum_{i=1}^n \mathbb{E}_{z_i'} \left[ \nabla f(A(S^{(i)}), z_i) \right] - \sum_{i=1}^n \nabla f(A(S), z_i),$$

which implies that

$$
\begin{aligned}
&\mathbb{E}_A \left\| n\nabla F(A(S)) - n\nabla F_S(A(S)) - \sum_{i=1}^{n} \mathbf{h}_i(S) \right\|_2 \\
&= \mathbb{E}_A \left\| \sum_{i=1}^{n} \mathbb{E}_Z \left[ \nabla f(A(S); Z) - \mathbb{E}_{z_i'} \left[ \nabla f(A(S^{(i)}), Z) \right] \right] \right. \\
&\qquad \left. + \sum_{i=1}^{n} \mathbb{E}_{z_i'} \left[ \nabla f(A(S^{(i)}), z_i) \right] - \sum_{i=1}^{n} \nabla f(A(S), z_i) \right\|_2 \\
&\leq 2n\beta,
\end{aligned}
\tag{17}
$$

where the inequality holds from the definition of uniform stability in gradients.

Then, for any $i = 1, \dots, n$, we define $\mathbf{q}_i(S) = \mathbf{h}_i(S) - \mathbb{E}_{S \setminus \{z_i\}, A}[\mathbf{h}_i(S)]$. It is easy to verify that $\mathbb{E}_{S \setminus \{z_i\}}[\mathbf{q}_i(S)] = \mathbf{0}$ and $\mathbb{E}_{z_i}[\mathbf{q}_i(S)] = \mathbb{E}_{z_i}[\mathbf{h}_i(S)] - \mathbb{E}_{z_i}\mathbb{E}_{S \setminus \{z_i\}, A}[\mathbf{h}_i(S)] = \mathbf{0} - \mathbf{0} = \mathbf{0}$. Also, for any $j \in [n]$ with $j \neq i$ and $z_j'' \in \mathcal{Z}$, we have the following inequality

$$
\begin{aligned}
&\mathbb{E}_A \left[ \| \mathbf{q}_i(S) - \mathbf{q}_i(z_1, \dots, z_{j-1}, z_j'', z_{j+1}, \dots, z_n) \|_2 \right] \\
&\leq \mathbb{E}_A \left[ \| \mathbf{h}_i(S) - \mathbf{h}_i(z_1, \dots, z_{j-1}, z_j'', z_{j+1}, \dots, z_n) \|_2 \right] \\
&\quad + \mathbb{E}_A \left[ \| \mathbb{E}_{S \setminus \{z_i\}}[\mathbf{h}_i(S)] - \mathbb{E}_{S \setminus \{z_i\}}[\mathbf{h}_i(z_1, \dots, z_{j-1}, z_j'', z_{j+1}, \dots, z_n)] \|_2 \right].
\end{aligned}
$$

For the first term $\mathbb{E}[\| \mathbf{h}_i(S) - \mathbf{h}_i(z_1, \dots, z_{j-1}, z_j'', z_{j+1}, \dots, z_n) \|_2]$, it can be bounded by $2\beta$ according to the definition of uniform stability. Similar result holds for the second term $\mathbb{E}[\| \mathbb{E}_{S \setminus \{z_i\}}[\mathbf{h}_i(S)] - \mathbb{E}_{S \setminus \{z_i\}}[\mathbf{h}_i(z_1, \dots, z_{j-1}, z_j'', z_{j+1}, \dots, z_n)] \|_2]$ according to the uniform stability. By a combination of the above analysis, we get $\mathbb{E}_A[\| \mathbf{q}_i(S) - \mathbf{q}_i(z_1, \dots, z_{j-1}, z_j'', z_{j+1}, \dots, z_n) \|_2] \leq 4\beta$.

Thus, we have verified that three conditions in Lemma 7 are satisfied for $\mathbf{q}_i(S)$. We have the following result for any $p \geq 2$

$$
\left\| \left\| \sum_{i=1}^{n} \mathbf{q}_i(S) \right\| \right\|_p \leq 2^{4 + \frac{1}{4}} \left( \sqrt{\frac{p}{e}} \right) (\sqrt{2p} + 1) n\beta \lceil \log_2 n \rceil.
$$

According to Lemma 4 for any $\delta \in (0, 1)$, with probability at least $1 - \delta/3$, we have

$$
\left\| \sum_{i=1}^{n} \mathbf{q}_i(S) \right\|_2 \leq 16 \times 2^{\frac{3}{4}} \sqrt{e} \beta \lceil \log_2 n \rceil \log (3e/\delta) + 16 \times 2^{\frac{1}{4}} \sqrt{e} \beta \lceil \log_2 n \rceil \sqrt{\log 3e/\delta}.
\tag{18}
$$

Furthermore, we can derive that

$$
\begin{aligned}
&n\nabla F(A(S)) - n\nabla F_S(A(S)) - \sum_{i=1}^{n} \mathbf{h}_i(S) + \sum_{i=1}^{n} \mathbf{q}_i(S) \\
&= n\nabla F(A(S)) - n\nabla F_S(A(S)) - \sum_{i=1}^{n} \mathbb{E}_{S \setminus \{z_i\}, A}[\mathbf{h}_i(S)] \\
&= n\nabla F(A(S)) - n\nabla F_S(A(S)) - n\mathbb{E}_{S', A}[\nabla F(A(S'))] + n\mathbb{E}_{S, A}[\nabla F(A(S))].
\end{aligned}
$$

Due to the i.i.d. property between $S$ and $S'$, we know that $\mathbb{E}_{S'}[\nabla F(A(S'))] = \mathbb{E}_S[\nabla F(A(S))]$. Thus, combined above equality, (17) and (18), we have

$$
\mathbb{E}_A \left[ \| n\nabla F(A(S)) - n\nabla F_S(A(S)) - n\mathbb{E}_{S,A}[\nabla F(A(S))] + n\mathbb{E}_{S',A}[\nabla F_S(A(S'))] \|_2 \right]
$$
$$
\leq \mathbb{E}_A \left[ \left\| n\nabla F(A(S)) - n\nabla F_S(A(S)) - \sum_{i=1}^n \mathbf{h}_i(S) \right\|_2 \right]
$$
$$
+ \mathbb{E}_A \left[ \left\| \sum_{i=1}^n \mathbf{h}_i(S) - n\mathbb{E}_{S,A}[\nabla F(A(S))] + n\mathbb{E}_{S',A} F_S[A(S')] \right\|_2 \right] \tag{19}
$$
$$
= \mathbb{E}_A \left[ \left\| n\nabla F(A(S)) - n\nabla F_S(A(S)) - \sum_{i=1}^n \mathbf{h}_i(S) \right\|_2 \right] + \mathbb{E}_A \left[ \left\| \sum_{i=1}^n \mathbf{q}_i(S) \right\|_2 \right]
$$
$$
\leq 2n\beta + 16 \times 2^{\frac{3}{4}} \sqrt{e} n\beta \lceil \log_2 n \rceil \log(3e/\delta) + 16 \times 2^{\frac{1}{4}} \sqrt{e} n\beta \lceil \log_2 n \rceil \sqrt{\log 3e/\delta}
$$
$$
\leq 16 \times 2^{\frac{3}{4}} \sqrt{e} n\beta \lceil \log_2 n \rceil \log(3e/\delta) + 32\sqrt{e} n\beta \lceil \log_2 n \rceil \sqrt{\log 3e/\delta}.
$$

This implies that for any $\delta \in (0,1)$, with probability at least $1 - \delta/3$, we have

$$
\mathbb{E}_A \left[ \| \nabla F(A(S)) - \nabla F_S(A(S)) \|_2 \right]
$$
$$
\leq \| \mathbb{E}_{S',A}[\nabla F_S(A(S'))] - \mathbb{E}_{S,A}[\nabla F(A(S))] \|_2 \tag{20}
$$
$$
+ 16 \times 2^{\frac{3}{4}} \sqrt{e}\beta \lceil \log_2 n \rceil \log(3e/\delta) + 32\sqrt{e}\beta \lceil \log_2 n \rceil \sqrt{\log 3e/\delta}.
$$

Next, we need to bound the term $\| \mathbb{E}_{S',A}[\nabla F_S(A(S'))] - \mathbb{E}_{S,A}[\nabla F(A(S))] \|_2$. There holds that $\| \mathbb{E}_{S,A}\mathbb{E}_{S'}[\nabla F_S(A(S'))] \|_2 = \| \mathbb{E}_{S,A}[\nabla F(A(S))] \|_2$. Then, by the Bernstein inequality in Lemma 5, we obtain the following inequality with probability at least $1 - \delta/3$,

$$
\left\| \mathbb{E}_{S',A}[\nabla F_S(A(S'))] - \mathbb{E}_{S,A}[\nabla F(A(S))] \right\|_2 \leq \sqrt{\frac{2\mathbb{E}_{z_i}[\| \mathbb{E}_{S',A}\nabla f(A(S'); z_i) \|_2^2] \log(6/\delta)}{n}} + \frac{M\log(6/\delta)}{n}. \tag{21}
$$

Then using Jensen's inequality, we have

$$
\mathbb{E}_{z_i}[\| \mathbb{E}_{S',A}\nabla f(A(S'); z_i) \|_2^2] \leq \mathbb{E}_{z_i}\mathbb{E}_{S',A}\| \nabla f(A(S'); z_i) \|_2^2
$$
$$
= \mathbb{E}_Z\mathbb{E}_{S',A}\| \nabla f(A(S'); Z) \|_2^2 = \mathbb{E}_Z\mathbb{E}_{S,A}\| \nabla f(A(S); Z) \|_2^2. \tag{22}
$$

Combing (20), (21) with (22), we finally obtain that with probability at least $1 - 2\delta/3$,

$$
\mathbb{E}_A \left[ \| \nabla F(A(S)) - \nabla F_S(A(S)) \|_2 \right]
$$
$$
\leq \sqrt{\frac{2\mathbb{E}_Z\mathbb{E}_{S,A}\| \nabla f(A(S); Z) \|_2^2 \log(6/\delta)}{n}} + \frac{M\log(6/\delta)}{n} \tag{23}
$$
$$
+ 16 \times 2^{\frac{3}{4}} \sqrt{e}\beta \lceil \log_2 n \rceil \log(3e/\delta) + 32\sqrt{e}\beta \lceil \log_2 n \rceil \sqrt{\log 3e/\delta}.
$$

Next, since $S = \{z_i, \ldots, z_n\}$, we define $p = p(z_1, \ldots, z_n) = \mathbb{E}_{Z,A}[\| \nabla f(A(S); Z) \|_2^2]$ and $p_i = p_i(z_1, \ldots, z_n) = \sup_{z_i \in \mathcal{Z}} p(z_i, \ldots, z_n)$. So there holds $p_i \geq p$ for any $i = 1, \ldots, n$ and any

$\{z_1, \ldots, z_n\} \in \mathcal{Z}^n$. Also, there holds that

$$\sum_{i=1}^{n}(p_i - p)^2$$

$$=\sum_{i=1}^{n}\left(\sup_{z_i \in \mathcal{Z}} \mathbb{E}_{Z,A}[\|\nabla f(A(S);Z)\|_2^2] - \mathbb{E}_{Z,A}[\|\nabla f(A(S);Z)\|_2^2]\right)^2$$

$$\leq\sum_{i=1}^{n}\left(\mathbb{E}_{Z,A}\left[\sup_{z_i \in \mathcal{Z}}\|\nabla f(A(S);Z)\|_2^2 - \|\nabla f(A(S);Z)\|_2^2\right]\right)^2$$

$$=\sum_{i=1}^{n}\left(\mathbb{E}_{Z,A}\left[\left(\sup_{z_i \in \mathcal{Z}}\|\nabla f(A(S);Z)\|_2 - \|\nabla f(A(S);Z)\|_2\right)\left(\sup_{z_i \in \mathcal{Z}}\|\nabla f(A(S);Z)\|_2 + \|\nabla f(A(S);Z)\|_2\right)\right]\right)^2$$

$$\leq\sum_{i=1}^{n}\beta^2\left(\mathbb{E}_{Z,A}\left[\|\nabla f(A(S);Z)\|_2 + \sup_{z_i \in \mathcal{Z}}\|\nabla f(A(S);Z)\|_2\right]\right)^2$$

$$\leq n\beta^2\left(2\mathbb{E}_{Z,A}[\|\nabla f(A(S);Z)\|_2 + \beta]\right)^2$$

$$\leq 8n\beta^2 p + 2n\beta^4,$$

(24)

where the first inequality follows from the Jensen's inequality. The second and third inequalities follow from the definition of uniform stability in gradients. The last inequality holds from that $(a+b)^2 \leq 2a^2 + 2b^2$.

From (24), we know that $p$ is $(8n\beta^2, 2n\beta^4)$ weakly self-bounded. Thus, by Lemma 6, we obtain that with probability at least $1 - \delta/3$,

$$\mathbb{E}_Z\mathbb{E}_{S,A}[\|\nabla f(A(S);Z)\|_2^2] - \mathbb{E}_{Z,A}[\|\nabla f(A(S);Z)\|_2^2]$$

$$\leq\sqrt{(16n\beta^2\mathbb{E}_S\mathbb{E}_Z[\|\nabla f(A(S);Z)\|_2^2] + 4n\beta^4)\log(3/\delta)}$$

$$=\sqrt{(\mathbb{E}_S\mathbb{E}_{Z,A}[\|\nabla f(A(S);Z)\|_2^2] + \frac{1}{4}\beta^2)16n\beta^2\log(3/\delta)}$$

$$\leq\frac{1}{2}(\mathbb{E}_S\mathbb{E}_{Z,A}[\|\nabla f(A(S);Z)\|_2^2] + \frac{1}{4}\beta^2) + 8n\beta^2\log(3/\delta),$$

where the last inequality follows from that $\sqrt{ab} \leq \frac{a+b}{2}$ for all $a, b > 0$. Thus, we have

$$\mathbb{E}_Z\mathbb{E}_{S,A}[\|\nabla f(A(S);Z)\|_2^2] \leq 2\mathbb{E}_{Z,A}[\|\nabla f(A(S);Z)\|_2^2] + \frac{1}{4}\beta^2 + 16n\beta^2\log(3/\delta). \quad (25)$$

Substituting (25) into (23), we finally obtain that with probability at least $1 - \delta$

$$\mathbb{E}_A\left[\|\nabla F(A(S)) - \nabla F_S(A(S))\|_2\right]$$

$$\leq\sqrt{\frac{2\left(2\mathbb{E}_{Z,A}[\|\nabla f(A(S);Z)\|_2^2] + \frac{1}{4}\beta^2 + 16n\beta^2\log(3/\delta)\right)\log(6/\delta)}{n}} + \frac{M\log(6/\delta)}{n} \quad (26)$$

$$+ 16 \times 2^{\frac{3}{4}}\sqrt{e}\beta\lceil\log_2 n\rceil\log(3e/\delta) + 32\sqrt{e}\beta\lceil\log_2 n\rceil\sqrt{\log 3e/\delta}.$$

According to inequality $\sqrt{a+b} = \sqrt{a} + \sqrt{b}$ for any $a, b > 0$, with probability at least $1 - \delta$, we have

$$\mathbb{E}_A\left[\|\nabla F(A(S)) - \nabla F_S(A(S))\|_2\right]$$

$$\leq\sqrt{\frac{4\mathbb{E}_{Z,A}[\|\nabla f(A(S);Z)\|_2^2]\log(6/\delta)}{n}} + \sqrt{\frac{\left(\frac{1}{2}\beta^2 + 32n\beta^2\log(3/\delta)\right)\log(6/\delta)}{n}} + \frac{M\log(6/\delta)}{n}$$

$$+ 16 \times 2^{\frac{3}{4}}\sqrt{e}\beta\lceil\log_2 n\rceil\log(3e/\delta) + 32\sqrt{e}\beta\lceil\log_2 n\rceil\sqrt{\log 3e/\delta}.$$

The proof is complete.

$\square$

*Proof of Proposition 1.* According to the proof in Theorem 2, we have the following inequality with probability at least $1 - \delta$

$$\mathbb{E}_A \left[\|\nabla F(A(S)) - \nabla F_S(A(S))\|_2\right]$$
$$\leq \sqrt{\frac{2\left(2\mathbb{E}_{Z,A}[\|\nabla f(A(S); Z)\|_2^2] + \frac{1}{4}\beta^2 + 16n\beta^2 \log(3/\delta)\right) \log(6/\delta)}{n}}$$
$$+ \frac{M \log(6/\delta)}{n} + 16 \times 2^{\frac{3}{4}} \sqrt{e}\beta \lceil \log_2 n \rceil \log(3e/\delta) + 32\sqrt{e}\beta \lceil \log_2 n \rceil \sqrt{\log 3e/\delta}. \tag{27}$$

Since SGC implies that $\mathbb{E}_Z[\|\nabla f(\mathbf{w}; Z)\|_2^2] \leq \lambda \|\nabla F(\mathbf{w})\|_2^2$, according to inequalities $\sqrt{ab} \leq \eta a + \frac{1}{\eta} b$ and $\sqrt{a + b} \leq \sqrt{a} + \sqrt{b}$ for any $a, b, \eta > 0$, we have the following inequality with probability at least $1 - \delta$

$$\mathbb{E}_A \left[\|\nabla F(A(S)) - \nabla F_S(A(S))\|_2\right]$$
$$\leq \sqrt{\frac{2\left(2\rho\mathbb{E}_A[\|\nabla F(A(S))\|_2^2] + \frac{1}{4}\beta^2 + 16n\beta^2 \log(3/\delta)\right) \log(6/\delta)}{n}}$$
$$+ \frac{M \log(6/\delta)}{n} + 16 \times 2^{\frac{3}{4}} \sqrt{e}\beta \lceil \log_2 n \rceil \log(3e/\delta) + 32\sqrt{e}\beta \lceil \log_2 n \rceil \sqrt{\log 3e/\delta}$$
$$\leq \sqrt{\frac{\left(\frac{1}{2}\beta^2 + 32n\beta^2 \log(3/\delta)\right) \log(6/\delta)}{n}} + \frac{\eta}{1 + \eta} \mathbb{E}_A[\|\nabla F(A(S))\|_2] + \frac{1 + \eta}{\eta} \frac{4\lambda M \log(6/\delta)}{n}$$
$$+ \frac{M \log(6/\delta)}{n} + 16 \times 2^{\frac{3}{4}} \sqrt{e}\beta \lceil \log_2 n \rceil \log(3e/\delta) + 32\sqrt{e}\beta \lceil \log_2 n \rceil \sqrt{\log 3e/\delta}.$$

which implies that

$$\mathbb{E}_A \left[\|\nabla F(A(S))\|_2\right] \leq (1 + \eta)\mathbb{E}_A \left[\|\nabla F_S(A(S))\|_2\right] + C\frac{1 + \eta}{\eta} \left(\frac{\lambda M}{n} \log(6/\delta) + \beta \log n \log(1/\delta)\right).$$

The proof is complete. □

*Proof of Remark 4.* According to the proof in Theorem 2, we have the following inequality that with probability at least $1 - \delta$

$$\mathbb{E}_A \left[\|\nabla F(A(S)) - \nabla F_S(A(S))\|_2\right]$$
$$\leq \sqrt{\frac{4\mathbb{E}_{Z,A}[\|\nabla f(A(S); Z)\|_2^2] \log(6/\delta)}{n}} + \sqrt{\frac{\left(\frac{1}{2}\beta^2 + 32n\beta^2 \log(3/\delta)\right) \log(6/\delta)}{n}} + \frac{M \log(6/\delta)}{n}$$
$$+ 16 \times 2^{\frac{3}{4}} \sqrt{e}\beta \lceil \log_2 n \rceil \log(3e/\delta) + 32\sqrt{e}\beta \lceil \log_2 n \rceil \sqrt{\log 3e/\delta}.$$
$$\tag{28}$$

Since $f(\mathbf{w})$ is $\gamma$-smooth, we have

$$\mathbb{E}_{Z,A}[\|\nabla f(A(S); Z)\|_2^2]$$
$$\leq \mathbb{E}_{Z,A}[\|\nabla f(A(S); Z) - \nabla f(\mathbf{w}^*; Z)\|_2^2 + \|\nabla f(\mathbf{w}^*; Z)\|_2^2] \tag{29}$$
$$\leq \gamma^2 \mathbb{E}_A \left[\|A(S) - \mathbf{w}^*\|_2^2\right] + \mathbb{E}_Z[\|\nabla f(\mathbf{w}^*; Z)\|_2^2]$$

Plugging (29) into (28), we have

$$\mathbb{E}_A\left[\|\nabla F(A(S)) - \nabla F_S(A(S))\|_2\right]$$

$$\leq \sqrt{\frac{4(\gamma^2 \mathbb{E}_A\left[\|A(S) - \mathbf{w}^*\|_2^2\right] + \mathbb{E}_Z[\|\nabla f(\mathbf{w}^*; Z)\|_2^2])\log(6/\delta)}{n}} + \sqrt{\frac{\left(\frac{1}{2}\beta^2 + 32n\beta^2 \log(3/\delta)\right)\log(6/\delta)}{n}}$$

$$+ \frac{M\log(6/\delta)}{n} + 16 \times 2^{\frac{3}{4}}\sqrt{e}\beta\left\lceil \log_2 n\right\rceil \log(3e/\delta) + 32\sqrt{e}\beta\left\lceil \log_2 n\right\rceil \sqrt{\log 3e/\delta}$$

$$\leq 2\gamma\sqrt{\frac{\mathbb{E}_A\left[\|A(S) - \mathbf{w}^*\|_2^2\right]\log(6/\delta)}{n}} + \sqrt{\frac{4\mathbb{E}_Z\left[\|\nabla f(\mathbf{w}^*; Z)\|_2^2\right]\log(6/\delta)}{n}}$$

$$+ \sqrt{\frac{\left(\frac{1}{2}\beta^2 + 32n\beta^2 \log(3/\delta)\right)\log(6/\delta)}{n}} + \frac{M\log(6/\delta)}{n}$$

$$+ 16 \times 2^{\frac{3}{4}}\sqrt{e}\beta\left\lceil \log_2 n\right\rceil \log(3e/\delta) + 32\sqrt{e}\beta\left\lceil \log_2 n\right\rceil \sqrt{\log 3e/\delta},$$

(30)

where the second inequality holds because $\sqrt{a+b} \leq \sqrt{a} + \sqrt{b}$ for any $a, b > 0$, which means that

$$\mathbb{E}_A[\|\nabla F(A(S)) - \nabla F_S(A(S))\|_2]$$

$$\lesssim \beta \log n \log(1/\delta) + \frac{\log(1/\delta)}{n} + \sqrt{\frac{\mathbb{E}_Z\left[\nabla\|f(\mathbf{w}^*; Z)\|_2^2\right]\log(1/\delta)}{n}} + \sqrt{\frac{\mathbb{E}_A\left[\|A(S) - \mathbf{w}^*\|_2^2\right]\log(1/\delta)}{n}}.$$

The proof is complete.

$\square$

### C.3 Proofs of Subsection 3.2

**Lemma 13.** *Assume for any $z$, $f(\cdot, z)$ is $M$-Lipschitz. If $A$ is $\beta$-uniformly-stable in gradients, then for any $\delta \in (0, 1)$, the following inequality holds with probability at least $1 - \delta$*

$$\mathbb{E}_A\left[\|\nabla F(A(S)) - \nabla F_S(A(S))\|_2^2\right]$$

$$\leq \sqrt{\frac{4\mathbb{E}_Z\left[\|\nabla f(A(S); Z)\|_2^2\right]\log(6/\delta)}{n}} + \sqrt{\frac{\left(\frac{1}{2}\beta^2 + 32n\beta^2 \log(3/\delta)\right)\log(6/\delta)}{n}}$$

$$+ \frac{M\log(6/\delta)}{n} + 16 \times 2^{\frac{3}{4}}\sqrt{e}\beta\left\lceil \log_2(n)\right\rceil \log(3e/\delta) + 32\sqrt{e}\beta\left\lceil \log_2(n)\right\rceil \sqrt{\log(3e/\delta)}.$$

*Proof of Lemma 13.* According to proof of Theorem 2, similar to (19), we have the following inequality.

$$\mathbb{E}_A\left[\|n\nabla F(A(S)) - n\nabla F_S(A(S)) - n\mathbb{E}_{S,A}[\nabla F(A(S))] + n\mathbb{E}_{S',A}[\nabla F_S(A(S'))]\|_2^2\right]$$

$$\leq 2\mathbb{E}_A\left[\left\|n\nabla F(A(S)) - n\nabla F_S(A(S)) - \sum_{i=1}^{n}\mathbf{h}_i(S)\right\|_2^2\right]$$

$$+ 2\mathbb{E}_A\left[\left\|\sum_{i=1}^{n}\mathbf{h}_i(S) - n\mathbb{E}_{S,A}[\nabla F(A(S))] + n\mathbb{E}_{S',A}F_S[A(S')]\right\|_2^2\right]$$

$$= 2\mathbb{E}_A\left[\left\|n\nabla F(A(S)) - n\nabla F_S(A(S)) - \sum_{i=1}^{n}\mathbf{h}_i(S)\right\|_2^2\right] + 2\mathbb{E}_A\left[\left\|\sum_{i=1}^{n}\mathbf{q}_i(S)\right\|_2^2\right]$$

$$\leq 8n^2\beta^2 + 1024\sqrt{2}en^2\beta^2\left(\lceil \log_2 n\rceil\right)^2 \log^2(3e/\delta) + 512\sqrt{2}en^2\beta^2\left(\lceil \log_2 n\rceil\right)^2 \log(3e/\delta)$$

$$\leq 8n^2\beta^2 + 2048\sqrt{2}en^2\beta^2\left(\lceil \log_2 n\rceil\right)^2 \log^2(3e/\delta),$$

where the second inequality follows from the definition of uniform stability in gradients and Cauchy-Bunyakovsky-Schwarz inequality.

This implies that for any $\delta \in (0, 1)$, with probability at least $1 - \delta/3$, we have

$$\mathbb{E}_A \left[ \|\nabla F(A(S)) - \nabla F_S(A(S))\|_2^2 \right]$$
$$\leq \|\mathbb{E}_{S',A}[\nabla F_S(A(S'))] - \mathbb{E}_{S,A}[\nabla F(A(S))]\|_2^2 + 8\beta^2 + 2048\sqrt{2}e\beta^2 \left(\lceil \log_2 n \rceil\right)^2 \log^2 (3e/\delta). \tag{31}$$

According to (21) (22) and (25), using Cauchy-Bunyakovsky-Schwarz inequality, for any $\delta \in (0, 1)$ with probability at least $1 - 2\delta/3$, we have

$$\|\mathbb{E}_{S',A}[\nabla F_S(A(S'))] - \mathbb{E}_{S,A}[\nabla F(A(S))]\|_2^2$$
$$\leq \frac{\left(8\mathbb{E}_{Z,A}[\|\nabla f(A(S); Z)\|_2^2] + \beta^2 + 64n\beta^2 \log(3/\delta)\right) \log(6/\delta)}{n} + \frac{2M^2 \log(6/\delta)}{n^2}.$$

Combing above inequality with (31), with probability at least $1 - \delta$, we have

$$\mathbb{E}_A \left[ \|\nabla F(A(S)) - \nabla F_S(A(S))\|_2^2 \right]$$
$$\leq \frac{\left(8\mathbb{E}_{Z,A}[\|\nabla f(A(S); Z)\|_2^2] + \beta^2 + 64n\beta^2 \log(3/\delta)\right) \log(6/\delta)}{n} + \frac{2M^2 \log(6/\delta)}{n^2}$$
$$+ 8\beta^2 + 2048\sqrt{2}e\beta^2 \left(\lceil \log_2 n \rceil\right)^2 \log^2 (3e/\delta).$$

The proof is complete.

$\square$

*Proof of Theorem 3.* Since $f(\mathbf{w})$ is $\gamma$-smooth, we have

$$\mathbb{E}_{Z,A}[\|\nabla f(A(S); Z)\|_2^2]$$
$$\leq \mathbb{E}_{Z,A}[\|\nabla f(A(S); Z) - \nabla f(\mathbf{w}^*; Z)\|_2^2 + \|\nabla f(\mathbf{w}^*; Z)\|_2^2] \tag{32}$$
$$\leq \gamma^2 \mathbb{E}_A \left[ \|A(S) - \mathbf{w}^*\|_2^2 \right] + \mathbb{E}_Z[\|\nabla f(\mathbf{w}^*; Z)\|_2^2]$$

Combing above inequality with Lemma 13, with probability least $1 - \delta$, we have

$$\mathbb{E}_A \left[ \|\nabla F(A(S)) - \nabla F_S(A(S))\|_2^2 \right]$$
$$\leq \frac{8\gamma^2 \mathbb{E}_A \left[ \|A(S) - \mathbf{w}^*\|_2^2 \right] \log(6/\delta)}{n} + \frac{8\mathbb{E}_Z \left[ \|\nabla f(\mathbf{w}^*; Z)\|_2^2 \right] \log(6/\delta)}{n}$$
$$+ 65\beta^2 \log(3/\delta) \log(6/\delta) + \frac{2M^2 \log(6/\delta)}{n^2} + 8\beta^2 + 2048\sqrt{2}e\beta^2 \left(\lceil \log_2 n \rceil\right)^2 \log^2 (3e/\delta). \tag{33}$$

When $F(\mathbf{w})$ satisfies the PL condition and $\mathbf{w}^*$ is the projection of $A(S)$ onto the solution set $\arg\min_{\mathbf{w} \in \mathcal{W}} F(\mathbf{w})$, there holds the following error bound property (refer to Theorem 2 in [19])

$$\|\nabla F(A(S))\|_2 \geq \mu \|A(S) - \mathbf{w}^*\|_2.$$

Thus, we have

$$\mu^2 \mathbb{E}_A \left[ \|A(S) - \mathbf{w}^*\|_2^2 \right] \leq \mathbb{E}_A \left[ \|\nabla F(A(S))\|_2^2 \right]$$
$$\leq 2\mathbb{E}_A \left[ \|\nabla F(A(S)) - \nabla F_S(A(S))\|_2^2 \right] + 2\mathbb{E}_A \left[ \|\nabla F_S(A(S))\|_2^2 \right]$$
$$\leq 2\mathbb{E}_A[\|\nabla F_S(A(S))\|_2^2] + \frac{16\gamma^2 \mathbb{E}_A \left[ \|A(S) - \mathbf{w}^*\|_2^2 \right] \log(6/\delta)}{n} + \frac{16\mathbb{E}_Z \left[ \|\nabla f(\mathbf{w}^*; Z)\|_2^2 \right] \log(6/\delta)}{n}$$
$$+ 130\beta^2 \log(3/\delta) \log(6/\delta) + \frac{4M^2 \log(6/\delta)}{n^2} + 16\beta^2 + 4096\sqrt{2}e\beta^2 \left(\lceil \log_2 n \rceil\right)^2 \log^2 (3e/\delta).$$

When $n \geq \frac{32\gamma^2 \log(6/\delta)}{\mu^2}$, we have $\frac{16\gamma^2 \log(6/\delta)}{n} \leq \frac{\mu^2}{2}$, then we can derive that

$$\mu^2 \mathbb{E}_A \left[ \|A(S) - \mathbf{w}^*\|_2^2 \right]$$
$$\leq 2\mathbb{E}_A[\|\nabla F_S(A(S))\|_2^2] + \frac{\mu^2}{2} \mathbb{E}_A \left[ \|A(S) - \mathbf{w}^*\|_2^2 \right] + \frac{16\mathbb{E}_Z \left[ \|\nabla f(\mathbf{w}^*; Z)\|_2^2 \right] \log(6/\delta)}{n}$$
$$+ 130\beta^2 \log(3/\delta) \log(6/\delta) + \frac{4M^2 \log(6/\delta)}{n^2} + 16\beta^2 + 4096\sqrt{2}e\beta^2 \left(\lceil \log_2 n \rceil\right)^2 \log^2 (3e/\delta).$$

This implies that
$$\mathbb{E}_A[\|A(S) - \mathbf{w}^*\|_2^2]$$
$$\leq \frac{2}{\mu^2}\Big(2\mathbb{E}_A[\|\nabla F_S(A(S))\|_2^2] + \frac{16\mathbb{E}_Z\left[\|\nabla f(\mathbf{w}^*; Z)\|_2^2\right]\log(6/\delta)}{n}$$
$$+ 130\beta^2 \log(3/\delta)\log(6/\delta) + \frac{4M^2\log(6/\delta)}{n^2} + 16\beta^2 + 4096\sqrt{2}e\beta^2 \left(\lceil\log_2 n\rceil\right)^2 \log^2\left(3e/\delta\right)\Big). \tag{34}$$

Then, substituting (34) into (33), when $n \geq \frac{32\gamma^2 \log(6/\delta)}{\mu^2}$, with probability at least $1 - \delta$
$$\mathbb{E}_A\left[\|\nabla F(A(S)) - \nabla F_S(A(S))\|_2^2\right]$$
$$\leq \mathbb{E}_A\left[\|\nabla F_S(A(S))\|_2^2\right] + + \frac{16\mathbb{E}_Z\left[\|\nabla f(\mathbf{w}^*; Z)\|_2^2\right]\log(6/\delta)}{n} + 130\beta^2 \log(3/\delta)\log(6/\delta) \tag{35}$$
$$+ \frac{4M^2\log(6/\delta)}{n^2} + 16\beta^2 + 4096\sqrt{2}e\beta^2 \left(\lceil\log_2 n\rceil\right)^2 \log^2\left(3e/\delta\right).$$

Since $F$ satisfies the PL condition with $\mu$, we have
$$\mathbb{E}_A[F(A(S))] - F(\mathbf{w}^*) \leq \frac{\mathbb{E}_A\left[\|\nabla F(A(S))\|_2^2\right]}{2\mu}, \quad \forall \mathbf{w} \in \mathcal{W}. \tag{36}$$

So to bound $\mathbb{E}_A[F(A(S))] - F(\mathbf{w}^*)$, we need to bound the term $\mathbb{E}_A\left[\|\nabla F(A(S))\|_2^2\right]$. And there holds
$$\mathbb{E}_A\left[\|\nabla F(A(S))\|_2^2\right] = 2\mathbb{E}_A\left[\|\nabla F(A(S)) - \nabla F_S(A(S))\|_2^2\right] + 2\mathbb{E}_A\left[\|\nabla F_S(A(S))\|_2^2\right]. \tag{37}$$

On the other hand, when $f$ is nonegative and $\gamma$-smooth, from Lemma 4.1 of [43], we have
$$\|\nabla f(\mathbf{w}^*; z)\|_2^2 \leq 4\gamma f(\mathbf{w}^*; z),$$
which implies that
$$\mathbb{E}_Z[\|\nabla f(\mathbf{w}^*; Z)\|_2^2] \leq 4\gamma\mathbb{E}_Z f(\mathbf{w}^*; Z) = 4\gamma F(\mathbf{w}^*). \tag{38}$$

Combing (35),(36), (37) and (38), using Cauchy-Bunyakovsky-Schwarz inequality, we can derive that
$$\mathbb{E}_A\left[F(A(S))\right] - F(\mathbf{w}^*)$$
$$\lesssim \frac{\mathbb{E}_A\left[\|\nabla F_S(A(S))\|_2^2\right]}{\mu} + \frac{\gamma F(\mathbf{w}^*)\log(1/\delta)}{\mu n} + \frac{M^2\log^2(1/\delta)}{\mu n^2} + \frac{\beta^2\log^2 n\log^2(1/\delta)}{\mu}.$$

The proof is complete.

$\square$

# D  Proofs of ERM

*Proof of Lemma 1.* Since $F_{S^{(i)}}(\mathbf{w}) = \frac{1}{n}\left(f(\mathbf{w}; z_i') + \sum_{j\neq i} f(\mathbf{w}, z_j)\right)$, we have
$$F_S(\hat{\mathbf{w}}^*(S^{(i)})) - F_S(\hat{\mathbf{w}}^*(S))$$
$$= \frac{f(\hat{\mathbf{w}}^*(S^{(i)}); z_i) - f(\hat{\mathbf{w}}^*(S); z_i)}{n} + \frac{\sum_{j\neq i}(f(\hat{\mathbf{w}}^*(S^{(i)}); z_j) - f(\hat{\mathbf{w}}^*(S); z_j))}{n}$$
$$= \frac{f(\hat{\mathbf{w}}^*(S^{(i)}); z_i) - f(\hat{\mathbf{w}}^*(S); z_i)}{n} + \frac{f(\hat{\mathbf{w}}^*(S); z_i') - f(\hat{\mathbf{w}}^*(S^{(i)}); z_i')}{n}$$
$$+ \left(F_{S^{(i)}}(\hat{\mathbf{w}}^*(S^{(i)})) - F_{S^{(i)}}(\hat{\mathbf{w}}^*(S))\right)$$
$$\leq \frac{f(\hat{\mathbf{w}}^*(S^{(i)}); z_i) - f(\hat{\mathbf{w}}^*(S); z_i)}{n} + \frac{f(\hat{\mathbf{w}}^*(S); z_i') - f(\hat{\mathbf{w}}^*(S^{(i)}); z_i')}{n}$$
$$\leq \frac{2M}{n}\|\hat{\mathbf{w}}^*(S^{(i)}) - \hat{\mathbf{w}}^*(S)\|_2,$$

where the first inequality follows from the fact that $\hat{\mathbf{w}}^*(S^{(i)})$ is the ERM of $F_{S^{(i)}}$ and the second inequality follows from the Lipschitz property. Furthermore, for $\hat{\mathbf{w}}^*(S^{(i)})$, the PL condition and smoothness property of $F_S$ imply that its closest optima point of $F_S$ is $\hat{\mathbf{w}}^*(S)$ (the global minimizer of $F_S$ is unique [50]). Then, $F_S$ satisfies the quadratic growth property [19], which means that

$$F_S(\hat{\mathbf{w}}^*(S^{(i)})) - F_S(\hat{\mathbf{w}}^*(S)) \geq \frac{\mu}{2}\|\hat{\mathbf{w}}^*(S^{(i)}) - \hat{\mathbf{w}}^*(S)\|_2^2.$$

Then we get

$$\frac{\mu}{2}\|\hat{\mathbf{w}}^*(S^{(i)}) - \hat{\mathbf{w}}^*(S)\|_2^2 \leq F_S(\hat{\mathbf{w}}^*(S^{(i)})) - F_S(\hat{\mathbf{w}}^*(S)) \leq \frac{2M}{n}\|\hat{\mathbf{w}}^*(S^{(i)}) - \hat{\mathbf{w}}^*(S)\|_2,$$

which implies that $\|\hat{\mathbf{w}}^*(S^{(i)}) - \hat{\mathbf{w}}^*(S)\|_2 \leq \frac{4M}{n\mu}$. Combined with the smoothness property of $f$ we obtain that for any $S^{(i)}$ and $S$

$$\forall z \in \mathcal{Z}, \quad \left\|\nabla f(\hat{\mathbf{w}}^*(S^{(i)}); z) - \nabla f(\hat{\mathbf{w}}^*(S); z)\right\|_2^2 \leq \frac{16M^2\gamma^2}{n^2\mu^2}.$$

The proof is complete. $\qquad\qquad\square$

*Proof of Theorem 4.* From Lemma 1, we have $\|\nabla f(\hat{\mathbf{w}}^*(S); z) - \nabla f(\hat{\mathbf{w}}^*(S'); z)\|_2 \leq \frac{4M\gamma}{n\mu}$. Since $\nabla F_S(\hat{\mathbf{w}}^*) = 0$, we have $\|\nabla F_S(\hat{\mathbf{w}}^*)\|_2 = 0$. According to Theorem 3, since ERM is a deterministic algorithm, we can derive that

$$F(\hat{\mathbf{w}}^*(S)) - F(\mathbf{w}^*) = \mathbb{E}[F(\hat{\mathbf{w}}^*(S))] - F(\mathbf{w}^*) \lesssim \frac{\gamma F(\mathbf{w}^*)\log(1/\delta)}{\mu n} + \frac{M^2\gamma^2\log^2 n\log^2(1/\delta)}{\mu^3 n^2}.$$

$\square$

# E  Proofs of PGD

*Proof of Theorem 5.* According to smoothness assumption and $\eta = 1/\gamma$, we can derive that

$$\begin{aligned}
&F_S(\mathbf{w}_{t+1}) - F_S(\mathbf{w}_t)\\
&\leq \langle \mathbf{w}_{t+1} - \mathbf{w}_t, \nabla F_S(\mathbf{w}_t)\rangle + \frac{\gamma}{2}\|\mathbf{w}_{t+1} - \mathbf{w}_t\|_2^2\\
&= -\eta_t\|\nabla F_S(\mathbf{w}_t)\|_2^2 + \frac{\gamma}{2}\eta_t^2\|\nabla F_S(\mathbf{w}_t)\|_2^2\\
&= \left(\frac{\gamma}{2}\eta_t^2 - \eta_t\right)\|\nabla F_S(\mathbf{w}_t)\|_2^2\\
&\leq -\frac{1}{2}\eta_t\|\nabla F_S(\mathbf{w}_t)\|_2^2.
\end{aligned}$$

According to above inequality and the assumptions that $F_S$ satisfies the PL condition with parameter $\mu$, we can prove that

$$F_S(\mathbf{w}_{t+1}) - F_S(\mathbf{w}_t) \leq -\frac{1}{2}\eta_t\|\nabla F_S(\mathbf{w}_t)\|_2^2 \leq -\mu\eta_t(F_S(\mathbf{w}_t) - F_S(\hat{\mathbf{w}}^*)),$$

which implies that

$$F_S(\mathbf{w}_{t+1}) - F_S(\hat{\mathbf{w}}^*) \leq (1 - \mu\eta_t)(F_S(\mathbf{w}_t) - F_S(\hat{\mathbf{w}}^*)).$$

According to the property for $\gamma$-smooth for $F_S$ and the PL condition property with parameter $\mu$ for $F_S$, we have

$$\frac{1}{2\gamma}\|\nabla F_S(\mathbf{w})\|_2^2 \leq F_S(\mathbf{w}) - F_S(\hat{\mathbf{w}}^*) \leq \frac{1}{2\mu}\|\nabla F_S(\mathbf{w})\|_2^2,$$

which means that $\frac{\mu}{\gamma} \leq 1$.

Then If $\eta_t = 1/\gamma$, $0 \leq 1 - \mu\eta_t < 1$, taking over $T$ iterations, we get

$$F_S(\mathbf{w}_{t+1}) - F_S(\hat{\mathbf{w}}^*) \leq (1 - \mu\eta_t)^T (F_S(\mathbf{w}_t) - F_S(\hat{\mathbf{w}}^*)). \tag{39}$$

Combined (39), the smoothness of $F_S$ and the nonnegative property of $f$, it can be derive that

$$\|\nabla F_S(\mathbf{w}_{T+1}))\|_2^2 = O\left((1 - \frac{\mu}{\gamma})^T\right).$$

On the other hand, from Lemma 2, we have $\|\nabla f(\mathbf{w}_{T+1}(S); z) - \nabla f(\mathbf{w}_{T+1}(S'); z)\|_2^2 \leq \frac{4M^2\gamma^2}{n^2\mu^2}$.
Since $\|\nabla F_S(\mathbf{w}_{T+1})\|_2 = O\left((1 - \frac{\mu}{\gamma})^T\right)$ and PGD is a deterministic algorithm, according to Theorem 3, we can derive that

$$F(\mathbf{w}_{T+1}) - F(\mathbf{w}^*) = \mathbb{E}[F(\mathbf{w}_{T+1})] - F(\mathbf{w}^*)$$
$$\lesssim \left(1 - \frac{\mu}{\gamma}\right)^{2T} + \frac{\gamma F(\mathbf{w}^*) \log(1/\delta)}{\mu n} + \frac{M^2\gamma^2 \log^2 n \log^2(1/\delta)}{\mu^3 n^2}.$$

Let $T \asymp \log n$, we have

$$F(\mathbf{w}_{T+1}) - F(\mathbf{w}^*) \lesssim \frac{\gamma F(\mathbf{w}^*) \log(1/\delta)}{\mu n} + \frac{M^2\gamma^2 \log^2 n \log^2(1/\delta)}{\mu^3 n^2}.$$

The proof is complete. □

## F  Proofs of SGD

We first introduce the necessary lemma for the optimization error bound.

**Lemma 14** ([19]). *Let $\{\mathbf{w}_t\}_t$ be the sequence produced by SGD with $\eta_t = \frac{2t+1}{2\mu(t+1)^2}$. Suppose Assumption 1 hold. Assume for all $z$, the function $\mathbf{w} \mapsto f(\mathbf{w}; z)$ is $M$-Lipschitz and $\gamma$-smooth and assume $F_S$ satisfies PL condition with parameter $\mu$. There holds that*

$$\mathbb{E}_A[F_S(\mathbf{w}_{T+1})] - F_S(\hat{\mathbf{w}}^*) \leq \frac{M^2\gamma}{2T\mu^2}.$$

*Proof of Lemma 3.* We have known that $F_{S^{(i)}}(\mathbf{w}) = \frac{1}{n}\left(f(\mathbf{w}; z_i') + \sum_{j \neq i} f(\mathbf{w}; z_j)\right)$. We denote $\hat{\mathbf{w}}^*(S^{(i)})$ be the ERM of $F_{S^{(i)}}(\mathbf{w})$ and $\hat{\mathbf{w}}_S^*$ be the ERM of $F_S(\mathbf{w})$. From Lemma 1, we know that

$$\forall z \in \mathcal{Z}, \quad \left\|\nabla f(\hat{\mathbf{w}}^*(S^{(i)}); z) - \nabla f(\hat{\mathbf{w}}^*(S); z)\right\|_2^2 \leq \frac{16M^2\gamma^2}{n^2\mu^2}.$$

Also, for $\mathbf{w}_t$, the PL condition property of $F_S$ implies that its closest optima point of $F_S$ is $\hat{\mathbf{w}}^*(S)$ (the global minimizer of $F_S$ is unique [50]). Then, there holds that

$$\frac{\mu}{2}\mathbb{E}_A\left[\|\mathbf{w}_t - \hat{\mathbf{w}}^*(S)\|_2^2\right] \leq \mathbb{E}_A[F_S(\mathbf{w}_t)] - F_S(\hat{\mathbf{w}}^*(S)) \leq \frac{M^2\gamma}{2t\mu^2}.$$

Thus we have $\mathbb{E}_A[\|\mathbf{w}_t - \hat{\mathbf{w}}^*(S)\|_2^2] \leq \frac{M^2\gamma}{t\mu^3}$. A similar relation holds between $\hat{\mathbf{w}}^*(S^{(i)})$ and $\mathbf{w}_t^i$. Combined with the Lipschitz property of $f$ we obtain that for $\forall z \in \mathcal{Z}$, there holds that

$$\mathbb{E}_A\left[\left\|\nabla f(\mathbf{w}_t; z) - \nabla f(\mathbf{w}_t^i; z)\right\|_2^2\right]$$
$$\leq 3\mathbb{E}_A\left[\|\nabla f(\mathbf{w}_t; z) - \nabla f(\hat{\mathbf{w}}^*(S); z)\|_2^2\right] + 3\left\|\nabla f(\hat{\mathbf{w}}^*(S); z) - \nabla f(\hat{\mathbf{w}}^*(S^{(i)}); z)\right\|_2^2$$
$$+ 3\mathbb{E}_A\left[\left\|\nabla f(\hat{\mathbf{w}}^*(S^{(i)}); z) - \nabla f(\mathbf{w}_t^i; z)\right\|_2^2\right]$$
$$\leq 3\gamma^2\mathbb{E}_A\left[\|\mathbf{w}_t - \hat{\mathbf{w}}^*(S)\|_2^2\right] + \frac{48M^2\gamma^2}{n^2\mu^2} + 3\gamma^2\mathbb{E}_A\left[\|\hat{\mathbf{w}}^*(S^{(i)}) - \mathbf{w}_t^i\|_2^2\right]$$
$$\leq \frac{6M^2\gamma^3}{t\mu^3} + \frac{48M^2\gamma^2}{n^2\mu^2}.$$

The proof is complete. □

*Proof of Theorem 6.* From Lemma 3, we have

$$\mathbb{E}_A\left[\left\|\nabla f(\mathbf{w}_{T+1};z) - \nabla f(\mathbf{w}_{T+1}^i;z)\right\|_2^2\right] \leq \frac{6M^2\gamma^3}{T\mu^3} + \frac{48M^2\gamma^2}{n^2\mu^2}. \tag{40}$$

On the other hand, according to the smoothness property of $F_S$ and Lemma 14, we have

$$\mathbb{E}_A\left[\|\nabla F_S(\mathbf{w}_{T+1})\|_2^2\right] \leq \frac{M^2\gamma^2}{T\mu^2}. \tag{41}$$

Using Theorem 3, with probability at least $1 - \delta$, we have

$$\mathbb{E}_A\left[F(\mathbf{w}_{T+1})\right] - F(\mathbf{w}^*)$$
$$\lesssim \frac{M^2\gamma^2}{T\mu^3} + \frac{\gamma F(\mathbf{w}^*)\log(1/\delta)}{\mu n} + \frac{M^2\log^2(1/\delta)}{\mu n^2} + \left(\frac{\gamma}{T\mu} + \frac{1}{n^2}\right)\frac{M^2\gamma^2\log^2 n\log^2(1/\delta)}{\mu^3}$$
$$\lesssim \frac{\gamma F(\mathbf{w}^*)\log(1/\delta)}{\mu n} + \left(\frac{\gamma}{T\mu} + \frac{1}{n^2}\right)\frac{M^2\gamma^2\log^2 n\log^2(1/\delta)}{\mu^3}.$$

Furthermore, choosing $T \asymp n^2$, we finally obtain that when $n \geq \frac{32\gamma^2\log(6/\delta)}{\mu^2}$, with probability at least $1 - \delta$

$$F(\mathbf{w}_{T+1}) - F(\mathbf{w}^*) \lesssim \frac{\gamma F(\mathbf{w}^*)\log(1/\delta)}{\mu n} + \frac{M^2\gamma^3\log^2 n\log^2(1/\delta)}{\mu^4 n^2}.$$

$\square$