# OpenReview forum: "Stability and Sharper Risk Bounds with Convergence Rate $\tilde{O}(1/n^2)$"
_NeurIPS.cc/2025/Conference — NeurIPS 2025 poster_

### Official Review · Reviewer_n1ME · 2025-06-29

**Clarity:** 3
**Significance:** 3
**Originality:** 3
**Rating:** 4
**Confidence:** 3

**Summary:**

The paper provides a method to show generalization bounds under uniform stability of gradients for methods optimizing Lipschitz, smooth and strongly convex functions. Starting with a generic generalization bound that replaces that dependence on the Lipschitz constant in prior work to expectation of squared length of the gradient, the paper then shows sharper bounds for ERM, PGD and SGD. Notably, in the low noise regime ($F(w^∗) <O(1/n)$), the bounds for ERM, PGD  is $\tilde{O}(1/n^2)$. For SGD in a single pass ($T=n$), the bound is $\tilde{O}(1/n)$ in high probability, and when $T=n^2$, the bound is $\tilde{O}(1/n^2)$, sharper than previously known results. Another result is a constant-level improvement to concentration of sums of vector valued functions.

**Questions:**

- Can the authors provide a comparison with prior works in terms of the dependence on the problem parameters (Lipschitzness, smoothness, strong convexity parameters)?

**Ethical Concerns:**

["NO or VERY MINOR ethics concerns only"]

**Final Justification:**

I maintain my score after the discussion phase.

**Limitations:**

Adequately addressed.

**Paper Formatting Concerns:**

No concern

**Quality:**

3

**Strengths And Weaknesses:**

**Strengths**
- I find the paper technically strong and all results are presented clearly. In particular, all results are presented in detailed comparison with prior works, which highlights where both the improvements and limitations are.
- The improvements in the bounds are meaningful and interesting. I think improving the generalization bound from $\tilde{O}(1/n)$ (limit of prior works) to $\tilde{O}(1/n^2)$ even under more restrictive assumptions is non-trivial.
- I don't find any big issues in the proofs except some minor typos.

**Weaknesses**
- As mentioned above, the main weakness of the paper comes from the fact that the analysis requires stronger assumptions than do prior works (eg, PL vs SC, requiring lower bounds for the sample size). Here, I think the authors should revise the language in the abstract ("under the same common assumptions"), which is not correct.
- Since the paper mainly focuses on the dependence on the sample size $n$, it is unclear how the bounds in terms of problem parameters are improved or degraded. Here a comparison with prior work will be helpful.

Minor:
- Sentence fragments in Line 337-339.
- Missing $C_p$ in Eq (7) Line 530.

---

> ### Author Rebuttal · Authors · 2025-07-31
>
> Thanks for your careful review. Here are the replies.
>
> > Weakness 1: As mentioned above, the main weakness of the paper comes from the fact that the analysis requires stronger assumptions than do prior works (eg, PL vs SC, requiring lower bounds for the sample size). Here, I think the authors should revise the language in the abstract ("under the same common assumptions"), which is not correct.
>
> Answer: Thanks for your feedback. We will revise the language in the final version to avoid any potential misunderstanding. We appreciate your valuable feedback.
>
> > Weakness 2: Since the paper mainly focuses on the dependence on the sample size $n$, it is unclear how the bounds in terms of problem parameters are improved or degraded. Here a comparison with prior work will be helpful.
>
> Answer: Thank you for your suggestion. We have updated the table to include Lipschitzness $M$, smoothness $\gamma$, and strong convexity parameters/PL condition $\mu$ and compared with existing works. We will use this revised table in the final version, as it presents comparisons with prior studies more clearly and straightforwardly.
>
>
> |Reference | Algorithm | Method | Assumptions | Iterations  | Sample Size | Bounds  |
> |:---:|:---:|:---:|:---:|:---:| :---:|:---:|
> | (Zhang et al., 2017) | ERM | UC |  Smooth, SC, LN | - | $ \Omega \left( \frac{\gamma^2 d}{\mu^2} \right) $ |  $ \tilde{O}\left(\frac{M^2}{\mu n^2}\right) $  |
> | (Xu and Zeevi, 2024) | ERM | UC  | Smooth, PL, LN | - |  $ \Omega \left( \frac{\gamma^2 d}{\mu^2} \right) $ |  $ \tilde{O} \left( \frac{M^2 + \mu^2}{\mu n^2} \right) $  |
> | (Xu and Zeevi, 2024) | PGD | UC  | Smooth, PL, LN | $T \asymp \log(n)$  |  $ \Omega \left( \frac{\gamma^2 d}{\mu^2} \right) $  |  $\tilde{O} \left( \frac{M^2 + \mu^2}{n^2} \right)$  |
> |(Li et al., 2021)| SGD  | UC  | Smooth, PL, LN | $T \asymp n^2$  |  $ \Omega \left( \frac{\gamma^2 d}{\mu^2} \right) $  |  $\tilde{O} \left( \frac{M^2 + \mu^2}{n^2} \right)$  |
> | (Li and Liu, 2021)  | SGD | AS  | Smooth, SC | $T \asymp n^2$  | -  |  $\tilde{O} \left( \frac{\gamma^2 M^2 }{\mu n} \right)$  |
> |(klochkov and Zhivotovskiy, 2021)| ERM | AS  | SC | -  |  -  |  $ \tilde{O} \left( \frac{M^2}{\mu n} \right) $  |
> |(klochkov and Zhivotovskiy, 2021)| PGD | AS  | Smooth,  SC | $T \asymp \log(n)$  |  -  |  $\tilde{O} \left( \frac{\gamma M^2}{\mu^2 n} \right)$  |
> |This work| ERM | AS  | Smooth, SC, LN | -  | $\Omega \left( \frac{\gamma^2}{\mu^2} \right) $ |  $\tilde{O} \left( \frac{M^2 \gamma^2}{\mu^3 n^2} \right)$   |
> |This work | PGD | AS  | Smooth, SC, LN | $T \asymp \log(n)$  |  $\Omega \left( \frac{\gamma^2}{\mu^2} \right) $  |   $\tilde{O} \left( \frac{M^2 \gamma^2}{\mu^3 n^2} \right)$ |
> |This work| SGD | AS  | Smooth, SC, LN | $T \asymp n^2$ |  $\Omega \left( \frac{\gamma^2}{\mu^2} \right) $  |  $\tilde{O} \left( \frac{M^2 \gamma^2}{\mu^3 n^2} \right)$ |
> |This work | SGD | AS  | Smooth, SC | $T \asymp n$ |  $\Omega \left( \frac{\gamma^2}{\mu^2} \right) $  |  $\tilde{O} \left( \frac{\gamma^2}{\mu n} + \frac{M^2 \gamma^2}{\mu^3 n^2} \right)$ |
>
>
>
> > Weakness 3: Minor.
>
> Answer: Thanks for noting these typos. We'll fix all minor errors in the final version to the best of our ability.
>
>
>
> > Question: Can the authors provide a comparison with prior works in terms of the dependence on the problem parameters (Lipschitzness, smoothness, strong convexity parameters)?
>
>
> Answer: Please refer to the answer for Weakness 2 and the table given above.
>
>
> **We’re happy to address any further questions you may have.**
>
> Reference
>
> L. Zhang, T. Yang, and R. Jin. Empirical risk minimization for stochastic convex optimization: $O(1/n)$-and $O(1/n^2)$-type of risk bounds. COLT 2017.
>
> Y. Xu and A. Zeevi. Towards optimal problem dependent generalization error bounds in statistical learning theory. Mathematics of Operations Research 2024.
>
> S. Li and Y. Liu. Improved learning rates for stochastic optimization: Two theoretical viewpoints.
>
> Y. Klochkov, and N. Zhivotovskiy. Stability and deviation optimal risk bounds with convergence rate $O(1/n)$. NeurIPS 2021.

---

> > ### Comment · Reviewer_n1ME · 2025-08-04
> >
> > Thank you for your response. Regarding the new table, can the authors comment on the changes in the dependence on the problem parameters. In particular, I see some extra factors incurred ($\mu$ for example). This seems to degrade the bound compared with prior results.

---

> > > ### Author Response · Authors · 2025-08-06
> > > **Reply to Reviewer n1ME**
> > >
> > > Thank you for your feedback. We will now address the comments regarding the changes in the dependence on the problem parameters. To facilitate a more intuitive comparison, we are re-presenting the table here.
> > >
> > > |Reference | Algorithm | Method | Assumptions | Iterations  | Sample Size | Bounds  |
> > > |:---:|:---:|:---:|:---:|:---:| :---:|:---:|
> > > | (Zhang et al., 2017) | ERM | UC |  Smooth, SC, LN | - | $ \Omega \left( \frac{\gamma^2 d}{\mu^2} \right) $ |  $ \tilde{O}\left(\frac{M^2}{\mu n^2}\right) $  |
> > > | (Xu and Zeevi, 2024) | ERM | UC | Smooth, PL, LN | - |  $ \Omega \left( \frac{\gamma^2 d}{\mu^2} \right) $ |  $ \tilde{O} \left( \frac{M^2 + \mu^2}{\mu n^2} \right) $  |
> > > | (Xu and Zeevi, 2024) | PGD | UC | Smooth, PL, LN | $T \asymp \log(n)$  |  $ \Omega \left( \frac{\gamma^2 d}{\mu^2} \right) $  |  $\tilde{O} \left( \frac{M^2 + \mu^2}{\mu n^2} \right)$  |
> > > |(Li and Liu, 2021)| SGD | UC | Smooth, PL, LN | $T \asymp n^2$ | $ \Omega \left( \frac{\gamma^2 d}{\mu^2} \right) $  |  $\tilde{O} \left( \frac{M^2 + \mu^2}{\mu n^2} \right)$  |
> > > | (Li and Liu, 2021)  | SGD | AS | Smooth, SC | $T \asymp n^2$  | -  | $\tilde{O} \left( \frac{\gamma^2 M^2 }{\mu n} \right)$  |
> > > |(klochkov and Zhivotovskiy, 2021)| ERM | AS | SC | -  |  -  |  $ \tilde{O} \left( \frac{M^2}{\mu n} \right) $  |
> > > |(klochkov and Zhivotovskiy, 2021)| PGD | AS | Smooth, SC | $T \asymp \log(n)$  |  -  |  $\tilde{O} \left( \frac{\gamma M^2}{\mu^2 n} \right)$  |
> > > |This work| ERM | AS | Smooth, SC, LN| -  | $\Omega \left( \frac{\gamma^2}{\mu^2} \right) $ |  $\tilde{O} \left( \frac{M^2 \gamma^2}{\mu^3 n^2} \right)$   |
> > > |This work | PGD | AS | Smooth, SC, LN| $T \asymp \log(n)$  |  $\Omega \left( \frac{\gamma^2}{\mu^2} \right) $ | $\tilde{O} \left( \frac{M^2 \gamma^2}{\mu^3 n^2} \right)$ |
> > > |This work | PGD | AS | Smooth, SC| $T \asymp \log(n)$  |  $\Omega \left( \frac{\gamma^2}{\mu^2} \right) $  | $\tilde{O} \left(  \frac{\gamma F(w^*) }{\mu n}  +  \frac{M^2 \gamma^2}{\mu^3 n^2} \right)$ |
> > > |This work| SGD | AS | Smooth, SC, LN| $T \asymp n^2$ |  $\Omega \left( \frac{\gamma^2}{\mu^2} \right) $  | $\tilde{O} \left( \frac{M^2 \gamma^2}{\mu^3 n^2} \right)$ |
> > > |This work | SGD | AS | Smooth, SC| $T \asymp n$ |  $\Omega \left( \frac{\gamma^2}{\mu^2} \right) $  | $\tilde{O} \left( \frac{\gamma F(w^*) }{\mu n} + \frac{M^2 \gamma^2}{\mu^3 n^2} \right)$ |
> > >
> > > **We wish to compare our results from the following five perspectives:**
> > >
> > > 1.  **Comparison with SGD results based on algorithmic stability (Li and Liu, 2021):** While both achieve an $O(1/n)$ bound, they require $T \asymp n^2$ and obtain an upper bound of $\frac{\gamma^2 M^2}{\mu n}$. In contrast, we only require $T \asymp n$ and achieve an upper bound of $\frac{\gamma F(w^\ast) }{\mu n}$. Clearly, our dependence on the parameters $\mu$ is consistent with theirs. For $\gamma$, we are better. More importantly, compared to their dependence on the gradient upper bound $M$, our bound depends on $F(w^\ast)$, the population risk at the optimal model.
> > >
> > > 2.  **Comparison with algorithmic stability results (Klochkov and Zhivotovskiy, 2021):** Focusing on their result for PGD, our results and discussions are similar to SGD. This improvement stems precisely from our use of Bernstein's inequality for the gradient generalization, yielding a tighter bound involving its variance.
> > >
> > > 3. **Further parameter dependence comparison:** Even directly comparing our $O(1/n^2)$ bound (achievable under stronger assumptions) with their $O(1/n)$ algorithmic stability bounds reveals the superiority of our result concerning parameter dependence. Consider the PGD bound from (Klochkov and Zhivotovskiy, 2021): for their bound to be meaningful (i.e., $\frac{\gamma M^2}{\mu^2 n} < 1$), its square must be smaller than itself ($ \frac{\gamma^2 M^4}{\mu^4 n^2} < \frac{\gamma M^2}{\mu^2 n} $). We observe that our bound $ \frac{M^2 \gamma^2}{\mu^3 n^2} $ exhibits a weaker parameter dependence than the square of their bound. In other words, when their bound is meaningful, our result is smaller even after considering the parameter dependencies.
> > >
> > > 4.  **Comparison regarding ERM:** We must acknowledge that compared to the algorithmic stability analysis for ERM in (Klochkov and Zhivotovskiy, 2021), which does not rely on a smoothness assumption, our assumptions are stronger. However, by incorporating additional smoothness assumptions, we can derive an $O(1/n^2)$ bound. This demonstrates that our approach offers a more refined framework for deriving tighter bounds using algorithmic stability.
> > >
> > > 5.  **Comparison with Uniform Convergence:** Compared to the $O(1/n^2)$ results via Uniform Convergence, we concede that although our bound is now independent of the dimension $d$, our dependence on $\mu$ and $\gamma$ is higher. It is indeed possible that in low-dimensional machine learning scenarios, bounds derived from uniform convergence might be tighter. This is a direction we aim to explore further in the future.
> > >
> > > **If anything remains unclear, we welcome further questions.**

---

> > > > ### Comment · Reviewer_n1ME · 2025-08-09
> > > >
> > > > Thank you for your reply. This cleared up my concerns. I maintain my inclination towards acceptance.

---

> > > > > ### Author Response · Authors · 2025-08-09
> > > > > **Thank you for your high evaluation of our submission!**
> > > > >
> > > > > Thank you for your valuable comments and engaging discussion. We appreciate the time and insight you've provided to strengthen our manuscript.
> > > > >
> > > > > We are pleased to have responded to your points and addressed your concerns. We will carefully incorporate your feedback and suggestions into the manuscript during revision.
> > > > >
> > > > > We remain available until the end of the discussion phase for any further questions or clarifications needed.
> > > > >
> > > > > Sincerely,
> > > > >
> > > > > The Authors

---

### Official Review · Reviewer_EPgb · 2025-07-01

**Clarity:** 2
**Significance:** 3
**Originality:** 3
**Rating:** 5
**Confidence:** 4

**Summary:**

In this paper, the authors introduce new high-probability excess risk upper bounds based on uniform stability in gradients. They first develop a general theoretical framework for deriving such bounds, under the assumptions of a Polyak–Łojasiewicz (PL) condition, smoothness, and Lipschitz continuity of the loss function. They then apply this framework to several learning algorithms, including Empirical Risk Minimization (ERM), Projected Gradient Descent (PGD), and Stochastic Gradient Descent (SGD). In particular, they show that in low-noise regimes, the excess risk bounds can achieve the fast rate of $\tilde{O}(1/n^2)$, notably without any dependence on the dimension $d$.

**Questions:**

1) Could you confirm that the upper bound in Definition 2 holds almost surely with respect to the randomness of the algorithm $A$? If so, this is not standard and actually corresponds to a stronger notion of stability than what is typically considered in the literature, where an expectation is usually taken with respect to the randomness of $A$ (see, e.g., [23]). I would appreciate it if the authors could clarify this point and clearly discuss this aspect in the main text.

2) The results in Section 3 are derived under the Polyak–Łojasiewicz (PL) condition, while in the applications, the objectives considered are strongly convex. Could the application part be extended to handle objectives that satisfy only the PL condition, without requiring strong convexity? Some discussion on this point would help clarify the generality of the framework.

3) I believe there may be a missing $M^2$ factor in the third term of the upper bound in Theorem 3. While $M^2$ can be regarded as a constant, it is not purely numerical and should likely appear explicitly in the bound. Could the authors clarify this point? Also, is it possible that the dimension $d$ is implicitly hidden in $M$? An explicit statement on this would be useful.

4) In Definition 3, is $\rho$ meant to be a constant? If so, I suggest using a different notation, as $\rho$ is already used to denote the distribution of $z$. Additionally, did you absorb the dependence on $\rho$ into $C$ in the proof of Proposition 1? As above, it would be clearer if these constants appeared explicitly in the final bounds.

**Ethical Concerns:**

["NO or VERY MINOR ethics concerns only"]

**Final Justification:**

The authors have addressed most of my concerns. I recommend acceptance, provided the promised revisions are made and the paper is carefully proofread to catch any remaining typos I may have missed.

**Limitations:**

### Limitations

- Please see my earlier comment regarding the specific notion of stability in gradients adopted in this paper. I am concerned that this almost-sure notion of stability might be more restrictive to establish for randomized optimization algorithms (beyond SGD), and could potentially lead to suboptimal bounds, especially in terms of the number of iterations $T$ required to achieve the stated rates. I would strongly encourage the authors to discuss this limitation explicitly. I am open to raising my overall score if this point is properly addressed.

- I believe Theorem 1 and Remark 2 could be moved to the appendix. This would free up space in the main text and allow the authors to expand on the proof sketch or main ideas behind Theorem 2, which currently deserve more emphasis.

- I recommend that the authors include precise cross-references to the proofs provided in the appendix. At present, the appendix sections are not explicitly cited in the main text, which makes it harder for readers to follow the technical arguments.

- It should be clearly stated in the notation section that the symbol $\lesssim$ is used to hide (numerical?) constant factors in the upper bounds, while $\mathcal{O}$ and $\tilde{\mathcal{O}}$ follow the Landau notation conventions for asymptotic comparisons (the latter hiding polylogarithmic factors). In machine learning papers, $\lesssim$ and $\mathcal{O}$ are sometimes used interchangeably, but here they have distinct meanings. Clarifying this would avoid confusion.

### Typos

- l.12 "algorithmic" $\rightarrow$ "algorithmic stability"?
- l.50 "and $\mathcal{O}(1/n)$"$\rightarrow$ "and terms in $\mathcal{O}(1/n)$"?
- l.120 "under" $\rightarrow$ ", the"
- l.164 "stable" $\rightarrow$ "stability"
- l.171 "$\gamma$-smoothness" $\rightarrow$ "$\gamma$-smooth"
- l.198  "even IF the algorithm"
- l.235-236 "can BE up to"?
- l.254 "analysis" $\rightarrow$ "analyze"
- Proof of Theorem 2 (l.630), first equality: there is an extra bracket "]". This is repeated several times in the proof.

**Quality:**

2

**Strengths And Weaknesses:**

### Strengths

Overall, the paper presents a very interesting, original, and tighter approach to deriving high-probability upper bounds on the excess risk of learning and optimization algorithms. I particularly appreciate that the authors focus on high-probability bounds, in contrast to the vast majority of stability-based generalization results that primarily address the expected excess risk. Moreover, the theoretical results are thoroughly analyzed, the optimality of the bounds is discussed, and the comparisons to related literature are well-executed and informative.

### Weaknesses

One weakness I see concerns the notion of stability in gradients that the authors adopt. From my understanding, the stability upper bound holds almost surely with respect to the randomness of the algorithm $A$, which is not standard and appears to be a significant limitation that deserves further discussion. Besides that, the paper is relatively dense, and the writing could be improved in places to enhance clarity and accessibility. For example, it is not entirely clear whether all results need to be stated in the main text (e.g., Theorem 1 does not seem essential and could be moved to the appendix). On the other hand, some of this saved space could be better used to provide more intuition or discussion about the proof techniques behind Theorems 2 and 3, which are the core contributions of the paper. Beyond these points, please see my detailed suggestions and questions below.

---

> ### Author Rebuttal · Authors · 2025-07-31
>
> Thank you for your thorough review. Here are the replies.
>
> > Weakness 1 \& Question 1 \& Limitation 1: Expectation is usually taken with respect to the randomness of $A$.
>
> Answer: This is a typo. Our Definition 2, consistent with the definition of uniformly-stable in gradients in (Lei. 2023), also needs to take expectation over the randomness of the algorithm $A$. For the ERM and PGD algorithms, since they are deterministic, this does not affect our results. For the proof regarding SGD, we need to add an expectation with respect to the randomness of the algorithm (since the noise introduced by the algorithm's inherent randomness is typically assumed to be independent of the dataset $S$). Our conclusion remains unchanged, meaning for Theorem 3, for any $\delta \in (0,1)$, when $n \geq \frac{16\gamma^2 \log\frac{6}{\delta}}{\mu^2}$, with probability at least $1-\delta$, we have:
> $$\mathbb{E}_A [F(A(S))] - F(w^{\ast}) \lesssim \frac{|| \mathbb{E}_A [\nabla F_S(A(S))]||
> _2^2}{\mu} + \frac{\gamma F(w^{\ast}) \log(1/\delta)}{\mu n} + \frac{M^2\log^2(1/\delta)}{\mu n^2} + \frac{\beta^2 \log^2 n \log^2(1/\delta)}{\mu}$$.
>
> For Theorem 7, for any $\delta \in (0,1)$, when $n \geq \frac{16\gamma^2 \log\frac{6}{\delta}}{\mu^2}$, with probability at least $1-\delta$, we have
> $$
> \mathbb{E}\_A [F(w\_{T+1})] -F(w^*) =  O \left(\frac{ \left\lceil \log\_2 (n) \right\rceil^2 \log(T) \log^5(1/\delta)}{T}\right)  + O \left(\frac{ \left\lceil \log\_2 (n) \right\rceil^2 \log^2(1/\delta)}{n^2}  +  \frac{ F ( w^{\ast} ) \log (1 / \delta ) }{n}  \right).
> $$
>
> For the training set $S$, we still only need to sample once, but we need to train multiple times through optimization algorithms to address algorithmic randomness.
>
> > Question 2: The results in Section 3 are derived under the Polyak–Łojasiewicz (PL) condition, while in the applications, the objectives considered are strongly convex. Could the application part be extended to handle objectives that satisfy only the PL condition, without requiring strong convexity? Some discussion on this point would help clarify the generality of the framework.
>
> Answer: The derivation of Theorem 3 requires the PL condition. Observing Theorem 3, we see that to achieve rates of $O(1/n^2)$, the algorithm's stability needs to reach $O(1/n)$. Based on (Hardt et al. 2016)'s derivation, achieving stability at this level requires the loss function to satisfy the Quadratic Growth condition, i.e., $\mu ||x - x^{\ast}||^2 \leq f(x) - f(x^*)$.
>
> In fact, if the function is weakly-convex (A function f : $\mathbb{R}^n \to \mathbb{R}$ is called $\lambda$-weakly convex if the function $f (x) + \frac{\lambda}{2}||x||^2$ is convex.), then the PL condition can lead to the Quadratic Growth condition (Liao et al. 2024).
>
> In this paper, we felt that introducing too many new assumptions might confuse readers. Considering that achieving rates of $O(1/n^2)$ is challenging, and the strongly convex assumption is also a common assumption in research on fast rates. Therefore, we did not elaborate in detail on this aspect. Thank you for your suggestion; we will include this discussion in the final version.
>
>
> > Question 3: I believe there may be a missing $M^2$ factor in the third term of the upper bound in Theorem 3. While $M^2$ can be regarded as a constant, it is not purely numerical and should likely appear explicitly in the bound. Could the authors clarify this point? Also, is it possible that the dimension $d$ is implicitly hidden in $M$? An explicit statement on this would be useful.
>
> Answer: Yes, the parameter $M$ will be explicitly displayed in the final version of the theoretical bound. The norm of the gradients may indeed have a potential relationship with the dimension d. However, on the one hand, practical optimization algorithms commonly employ gradient clipping to prevent gradient explosion; on the other hand, existing works also include an $M$ of comparable magnitude in their bounds. We have updated the table to include Lipschitzness $M$, smoothness $\gamma$, and strong convexity parameters/PL condition $\mu$ and compared with existing works. Thus, we did not incorporate the dimension $d$ into $M$ implicitly as a means to achieve theoretical improvements.
>
>
>
>
> |Reference | Algorithm | Method | Assumptions | Iterations  | Sample Size | Bounds  |
> |:---:|:---:|:---:|:---:|:---:| :---:|:---:|
> | (Zhang et al., 2017) | ERM | UC |  Smooth, SC, LN | - | $ \Omega \left( \frac{\gamma^2 d}{\mu^2} \right) $ |  $ \tilde{O}\left(\frac{M^2}{\mu n^2}\right) $  |
> | (Xu and Zeevi, 2024) | ERM | UC  | Smooth, PL, LN | - |  $ \Omega \left( \frac{\gamma^2 d}{\mu^2} \right) $ |  $ \tilde{O} \left( \frac{M^2 + \mu^2}{\mu n^2} \right) $  |
> | (Xu and Zeevi, 2024) | PGD | UC  | Smooth, PL, LN | $T \asymp \log(n)$  |  $ \Omega \left( \frac{\gamma^2 d}{\mu^2} \right) $  |  $\tilde{O} \left( \frac{M^2 + \mu^2}{n^2} \right)$  |
> |(Li et al., 2021)| SGD  | UC  | Smooth, PL, LN | $T \asymp n^2$  |  $ \Omega \left( \frac{\gamma^2 d}{\mu^2} \right) $  |  $\tilde{O} \left( \frac{M^2 + \mu^2}{n^2} \right)$  |
> | (Li and Liu, 2021)  | SGD | AS  | Smooth, SC | $T \asymp n^2$  | -  |  $\tilde{O} \left( \frac{\gamma^2 M^2 }{\mu n} \right)$  |
> |(klochkov and Zhivotovskiy, 2021)| ERM | AS  | SC | -  |  -  |  $ \tilde{O} \left( \frac{M^2}{\mu n} \right) $  |
> |(klochkov and Zhivotovskiy, 2021)| PGD | AS  | Smooth,  SC | $T \asymp \log(n)$  |  -  |  $\tilde{O} \left( \frac{\gamma M^2}{\mu^2 n} \right)$  |
> |This work| ERM | AS  | Smooth, SC, LN | -  | $\Omega \left( \frac{\gamma^2}{\mu^2} \right) $ |  $\tilde{O} \left( \frac{M^2 \gamma^2}{\mu^3 n^2} \right)$   |
> |This work | PGD | AS  | Smooth, SC, LN | $T \asymp \log(n)$  |  $\Omega \left( \frac{\gamma^2}{\mu^2} \right) $  |   $\tilde{O} \left( \frac{M^2 \gamma^2}{\mu^3 n^2} \right)$ |
> |This work| SGD | AS  | Smooth, SC, LN | $T \asymp n^2$ |  $\Omega \left( \frac{\gamma^2}{\mu^2} \right) $  |  $\tilde{O} \left( \frac{M^2 \gamma^2}{\mu^3 n^2} \right)$ |
> |This work | SGD | AS  | Smooth, SC | $T \asymp n$ |  $\Omega \left( \frac{\gamma^2}{\mu^2} \right) $  |  $\tilde{O} \left( \frac{\gamma^2}{\mu n} + \frac{M^2 \gamma^2}{\mu^3 n^2} \right)$ |
>
>
>
>
>
> > Question 4: In Definition 3, is $\rho$ meant to be a constant? If so, I suggest using a different notation, as $\rho$ is already used to denote the distribution of $z$. Additionally, did you absorb the dependence on $\rho$ into $C$ in the proof of Proposition 1? As above, it would be clearer if these constants appeared explicitly in the final bounds.
>
> Answer: Thanks for catching these points. We will select different symbols to avoid confusion. We'll also revise the bound to show constants depending on $\rho$ in the final version.
>
>
>
> > Weakness 2 \& Limitation 2  \& Limitation 3 \& Limitation 4: Writing suggestions.
>
> Answer: We appreciate your feedback and will implement your suggestions in the final version to the best of our ability, enhancing the paper's readability and eliminating ambiguities.
>
> > Limitation 5: Typos.
>
> Answer: Thanks for noting these typos. We'll fix all minor errors in the final version to the best of our ability.
>
>
> **We’re happy to address any further questions you may have.**
>
> Reference
>
> Moritz Hardt et al. Train faster, generalize better: Stability of stochastic gradient descent. 2016.
>
> F. Liao et al. Error bounds, PL condition, and quadratic growth for weakly convex functions, and linear convergences of proximal point methods. 2024.
>
> Y. Lei. Stability and generalization of stochastic optimization with nonconvex and nonsmooth problems. COLT 2023.
>
> L. Zhang, T. Yang, and R. Jin. Empirical risk minimization for stochastic convex optimization: $O(1/n)$-and $O(1/n^2)$-type of risk bounds. COLT 2017.
>
> Y. Xu and A. Zeevi. Towards optimal problem dependent generalization error bounds in statistical learning theory. Mathematics of Operations Research 2024.
>
> S. Li and Y. Liu. Improved learning rates for stochastic optimization: Two theoretical viewpoints.
>
> Y. Klochkov, and N. Zhivotovskiy. Stability and deviation optimal risk bounds with convergence rate $O(1/n)$. NeurIPS 2021.

---

> > ### Comment · Reviewer_EPgb · 2025-08-04
> >
> > I would like to thank the authors for their response, which addressed most of my concerns.

---

> > > ### Author Response · Authors · 2025-08-07
> > > **Thank you for your high evaluation of our submission!**
> > >
> > > Thank you for your valuable comments and engaging discussion. We appreciate the time and insight you've provided to strengthen our manuscript.
> > >
> > > We are pleased to have responded to your points and addressed your concerns. We will carefully incorporate your feedback and suggestions into the manuscript during revision.
> > >
> > > We remain available until the end of the discussion phase for any further questions or clarifications needed.
> > >
> > > Sincerely,
> > > The Authors

---

### Official Review · Reviewer_D3wC · 2025-07-02

**Clarity:** 3
**Significance:** 3
**Originality:** 3
**Rating:** 5
**Confidence:** 3

**Summary:**

This paper focuses on the question whether tighter high-probability excess risk bound tighter than $O(1/n)$ can be obtained using algorithmic stability-based methods and under common assumptions (e.g., strong convexity, smoothness, Lipschitz, PL ...).
To answer this question, an dimension-free $\tilde{O}(1/n^2)$ bounds are derived under Lipschitzness, smoothness and PL condition.
A dimension-free improved gradient generalization is first proved to potentially allow $O(1/n)$ rates under suitable stability and regularity. Then combined with PL condition, the generalization of gradient is translated into the generalization of model weight, where the square in the PL condition promotes the rate to potentially $O(1/n^2)$ under suitable stability. Finally, the stability bounds are applied to ERM, PGD and SGD by proving plugging-in their stability.

**Questions:**

1. Is it possible to decrease the order of $\log \frac{1}{\delta}$ terms in Theorems 6 and 7?
2. The following questions may be not necessary. It would be greatly appreciated if the authors could satisfy our curiosity. In [19], an $\tilde{O}(1/n)$ bound is established under strong convexity and Lipschitzness, and in this paper an $\tilde{O}(1/n^2)$ bound is established under strong convexity, Lipschitzness and Smoothness, which can be seen as the Lipschitzness of gradients or "first-order Lipschitzness". Therefore, is it possible that given strong convexity and $0-, 1-, \dots p-$th order Lipschitzness (perhaps with more minor assumptions), an $\tilde{O}(1/n^{p+1})$ convergence can be established. It is particularly interesting after realizing that given high-order Lipschitzness (e.g., 2-order Lipschitzness on Hessian) and extra assumptions of boundedness of hypothesis space, one can prove lower-order Lipschitzness by integral and obtain lower-order (at least 2-order) convergence. Then the only remained question is whether the high-order Lipschitzness can be "fully" exploited to achieve $(p+1)$-order convergence.

**Ethical Concerns:**

["NO or VERY MINOR ethics concerns only"]

**Final Justification:**

The authors have addressed all my concerns, and I keep my rating.

**Limitations:**

yes

**Paper Formatting Concerns:**

No.

**Quality:**

3

**Strengths And Weaknesses:**

## Strengths

1. This paper derives the first dimension-free $\tilde{O}(1/n^2)$ bound on excess risk under commonly assumed strong convexity/PL, smoothness, Lipschitzness. The results are carefully compared (e.g., Table 1 and the remarks after each Theorems) and are optimal under some conditions, demonstrating the superiority in convergence rate or dimension dependence.
2. The idea of using PL condition to bring a square in stability bounds and accelerate its rate is quite novel.
3. The contribution is not limited to generalization of risks, but also the generalization w.r.t. gradients. The new generalization bound w.r.t. gradient has the same convergence rage but has dimension-free requirements on $n$ and also applies to non-convex and non-smooth cases under Lipschitzness and SGC.
4. The paper is clearly structured and well written.

## Weaknesses


1. For SGD results to have $O(1/n^2)$ rate, the step number has to be $T = n^2$. Moreover, Theorems 6 and 7 have $\log^{\{3, 5\}} \frac{1}{\delta}$ terms of very high order.
2. Aside from the use of PL condition and the consequential order improvement in convergence rate, the technique to achieve them is based on Klochkov and 147 Zhivotovskiy [19]. Nevertheless, I believe this is minor compared to the $\tilde{O}(1/n^2)$ rate.
3. Minor/Typos:
   1. The sentence in L267-L269 is quite confusing: "However, the generalization analysis method proposed by Klochkov and Zhivotovskiy [19] does not use smoothness assumption, which only derive ... under strongly convex and smooth assumptions."

---

> ### Author Rebuttal · Authors · 2025-07-31
>
> Thank you for your thorough review. Here are the replies.
>
> > Weakness 1 \& Question 1: Is it possible to decrease the order of $\log\frac{1}{\delta}$ terms in Theorems 6 and 7?
>
>
> Answer: After careful consideration, we acknowledge that the current analysis does not readily support a further reduction in the logarithmic order of these terms. While we recognize the theoretical importance of optimizing such bounds, our present framework relies on concentration inequalities and union-bound arguments that inherently introduce this logarithmic dependency. We agree that this is an important direction for future work, and we are actively exploring alternative techniques that might mitigate these terms.
>
>
> > Weakness 2: Aside from the use of PL condition and the consequential order improvement in convergence rate, the technique to achieve them is based on Klochkov and 147 Zhivotovskiy [19]. Nevertheless, I believe this is minor compared to the $\tilde{O}(1/n^2)$ rate.
>
> Answer: Thank you for acknowledging our work. Although our proof approach was inspired by theirs, we encountered significant challenges. The existing method is fundamentally built upon a one-dimensional framework, while ours operates in a high-dimensional regime. Their technique cannot be generalized to our case.
>
>
> > Weakness 3: Minor/Typos.
>
> Answer: Thanks for noting these typos. We'll fix all minor errors in the final version to the best of our ability.
>
> > Weakness 4: The sentence in L267-L269 is quite confusing: "However, the generalization analysis method proposed by Klochkov and Zhivotovskiy [19] does not use smoothness assumption, which only derive ... under strongly convex and smooth assumptions."
>
> Answer: This statement suggests that Klochkov and Zhivotovskiy [19] did not employ the smoothness assumption. Instead, under the same number of iterations, their use of the PGD algorithm can only achieve $O(1/n)$ rates. Thanks for your advice. We will thoroughly polish the paper in the final version.
>
>
>
> > Question 2: The following questions may be not necessary. It would be greatly appreciated if the authors could satisfy our curiosity...
>
> Answer: The core idea of our approach rests on performing an expansion at  first-order stationary point where the gradient vanishes. The point where we perform this expansion is itself sufficiently small, enabling higher-order generalization. However, if we attempt an analogous expansion involving higher-order terms like the Hessian, the Hessian at that point is not guaranteed to be sufficiently small, preventing us from achieving tighter generalization bounds.
>
>
> **We’re happy to address any further questions you may have.**
>
> Reference
>
> Yegor Klochkov, and Nikita Zhivotovskiy. Stability and deviation optimal risk bounds with convergence rate $O(1/n)$. NeurIPS 2021.

---

### Official Review · Reviewer_t9CR · 2025-07-03

**Clarity:** 3
**Significance:** 2
**Originality:** 2
**Rating:** 4
**Confidence:** 3

**Summary:**

This paper presents high-probability excess risk bounds for empirical risk minimization (ERM) problems and projected gradient descent (PGD)/ stochastic gradient descent (SGD) algorithms. The authors achieve dimension-free, O(1/n) generalization bounds by generalizing the standard
concept of algorithmic stability of loss function to its gradient. This approach allows for sharper generalization bound.

This work also combines the derived generalization bound on gradient and analysis on optimization error for strongly convexity or Polyak-Lojasiewicz (PL) condition and implies a sharp bound on the excess risk.

**Questions:**

Comment: The authors should specify what you mean by PGD. I thought it was GD but I have to find the original reference to know it was projected GD.

**Ethical Concerns:**

["NO or VERY MINOR ethics concerns only"]

**Final Justification:**

Thanks for the author rebuttal.

I updated my review.

**Quality:**

3

**Strengths And Weaknesses:**

Strengths
- The generation bounds were developed before, but the one in this paper is dimension free.
- The results are well-presented.

Weaknesses
- The technical novelty is somewhat limited, with proofs heavily relying on appropriate combinations or refinements of existing techniques (e.g., Zhang & Zhou 2019, Klochkov & Zhivotovskiy 2021, Fan & Lei 2024).
- In the strongly convex/ PL setting, if $\mu$ is small, the bound is not good as it has $1/\mu^3$
- The assumption $n \geq \frac{16\gamma^2 \log(6/\delta)}{\mu^2}$ and $F(w*) < O (1/n)$ might not be practical in the strongly convex applications.

---

> ### Author Rebuttal · Authors · 2025-07-31
>
> Thanks for your review. Here are the replies.
>
> > Strength 1: The generation bounds were developed before, but the one in this paper is dimension free.
>
>
> Answer: We disagree with your claim presented in this paper that "generalization bounds were developed before". High-probability stability-based excess risk upper bounds achieving $O(1/n^2)$ have never been presented before; the best existing result only achieves $O(1/n)$.
>
> Furthermore, the stability-based generalization error bound for the gradients we derived is also tightest known. By utilizing the vector Bernstein inequality, the constant coefficient preceding the $\sqrt{1/n}$ term now depends not on the maximum gradient norm $M$, but on a coefficient related to the variance. Critically, this conclusion also holds (is applicable) in non-convex settings.
>
>
>
>
>
> > Weakness 1: The technical novelty is somewhat limited, with proofs heavily relying on appropriate combinations or refinements of existing techniques (e.g., Zhang \& Zhou 2019, Klochkov \& Zhivotovskiy 2021, Fan \& Lei 2024).
>
> Answer: We respectfully disagree with the assessment of limited novelty.  We summarize the difference with the mentioned literature.
>
> 1. **(Zhang \& Zhou, 2019) studied optimization problems, whereas we study generalization problems.** I guess you intended to cite the paper (Zhang et al., 2017) here.
>
>
> 2. **Totally different techniques to (Zhang et al. 2017)**: They use the uniform convergence techniques to get generalization bounds which dependent on dimension $d$. As the uniform convergence method inevitably introduce a dimension-dependent bound, we **improve this** with a **totally different** techniques, **the gradient stability**, and get our dimension-free $O(1 / n^{2})$ bound.
>
> 3. **From function stability to gradient stability**. Compared to (Klochkov \& Zhivotovskiy 2021), they prove their high probability generalization bound via function stability (Definition 6 in (Bousquet and Elisseeff, 2002)), which typically implies only a $O(1/n)$ bound. We improve this by introducing gradient stability (Definition 2 in this paper), which leads to the $O(1/n^2)$ generalization bound demonstrated in this paper.
>
>
> 4. We sharpen the fundamental gradient stability bound via a **novel Bernstein's inequality** (Theorem 2): In (Fan \& Lei 2024), they built the gradient stability (we show this in this paper in Theorem 1), similar to our Theorem 2. But the critical difference is that constants in $1/\sqrt{n}$ is improved from $O\left(M \sqrt{\log{(e/\delta)}} \right)$ to $ O \left( \sqrt{\mathbb{E}\_{Z} \left[ || \nabla f(A(S);Z) ||^2 \right] \log{(1/\delta)}} \right)$, since $M$ is the maximal gradient norm. Notably, the proved dependence $\mathbb{E}_{Z} \left[||\nabla f(A(S); Z) ||_2^{2} \right]$ via novel Bernstein's inequality is interpreted as the squared gradient norm of the loss functions on test data, under the optimized model parameters $A(S)$. Thus, it is supposed to be small when the algorithm is good, compared with constant $M$.
>
>
>
>
>
> > Weakness 2: In the strongly convex/ PL setting, if $\mu$ is small, the bound is not good as it has $1/\mu^3$.
>
> Answer:
> 1. The $\mu$ is not necessary to be small: We only require the population risk $F$ to satisfies PL condition (weaker than the PL condition for empirical risk $F_S$) in Theorem 3, and we verify it is $2$ under a linear regression example in Remark 5.
>
> 2. **The dependence on $\mu$ is standard**: Existing works have also dependence on $\mu$ as presented in the following table.
>
>
> |Reference | Algorithm | Method | Assumptions | Iterations  | Sample Size | Bounds  |
> |:---:|:---:|:---:|:---:|:---:| :---:|:---:|
> | (Zhang et al., 2017) | ERM | UC |  Smooth, SC, LN | - | $ \Omega \left( \frac{\gamma^2 d}{\mu^2} \right) $ |  $ \tilde{O}\left(\frac{M^2}{\mu n^2}\right) $  |
> | (Xu and Zeevi, 2024) | ERM | UC  | Smooth, PL, LN | - |  $ \Omega \left( \frac{\gamma^2 d}{\mu^2} \right) $ |  $ \tilde{O} \left( \frac{M^2 + \mu^2}{\mu n^2} \right) $  |
> | (Xu and Zeevi, 2024) | PGD | UC  | Smooth, PL, LN | $T \asymp \log(n)$  |  $ \Omega \left( \frac{\gamma^2 d}{\mu^2} \right) $  |  $\tilde{O} \left( \frac{M^2 + \mu^2}{n^2} \right)$  |
> |(Li et al., 2021)| SGD  | UC  | Smooth, PL, LN | $T \asymp n^2$  |  $ \Omega \left( \frac{\gamma^2 d}{\mu^2} \right) $  |  $\tilde{O} \left( \frac{M^2 + \mu^2}{n^2} \right)$  |
> | (Li and Liu, 2021)  | SGD | AS  | Smooth, SC | $T \asymp n^2$  | -  |  $\tilde{O} \left( \frac{\gamma^2 M^2 }{\mu n} \right)$  |
> |(klochkov and Zhivotovskiy, 2021)| ERM | AS  | SC | -  |  -  |  $ \tilde{O} \left( \frac{M^2}{\mu n} \right) $  |
> |(klochkov and Zhivotovskiy, 2021)| PGD | AS  | Smooth,  SC | $T \asymp \log(n)$  |  -  |  $\tilde{O} \left( \frac{\gamma M^2}{\mu^2 n} \right)$  |
> |This work| ERM | AS  | Smooth, SC, LN | -  | $\Omega \left( \frac{\gamma^2}{\mu^2} \right) $ |  $\tilde{O} \left( \frac{M^2 \gamma^2}{\mu^3 n^2} \right)$   |
> |This work | PGD | AS  | Smooth, SC, LN | $T \asymp \log(n)$  |  $\Omega \left( \frac{\gamma^2}{\mu^2} \right) $  |   $\tilde{O} \left( \frac{M^2 \gamma^2}{\mu^3 n^2} \right)$ |
> |This work| SGD | AS  | Smooth, SC, LN | $T \asymp n^2$ |  $\Omega \left( \frac{\gamma^2}{\mu^2} \right) $  |  $\tilde{O} \left( \frac{M^2 \gamma^2}{\mu^3 n^2} \right)$ |
> |This work | SGD | AS  | Smooth, SC | $T \asymp n$ |  $\Omega \left( \frac{\gamma^2}{\mu^2} \right) $  |  $\tilde{O} \left( \frac{\gamma^2}{\mu n} + \frac{M^2 \gamma^2}{\mu^3 n^2} \right)$ |
>
>
>
>
> > Weakness 3: The assumption $n\geq\frac{16\gamma^2\log(6/\delta)}{\mu^2}$  and $F(w^*)<O(1/n)$ might not be practical in the strongly convex applications.
>
> Answer:
>
> 1. For the condition on $n$, this is a standard condition to expect the machine learning model **can generalize**: In general, when consider generalization bound, the sample size is expected goes to infinity. Our assumption only requires the sample size is larger than a constant size, if the condition is not satisfied, none of existing literature obtain a small generalization bound.
>
>
> 2. In many cases, the optimal population risk $F(w^{\ast})$ can be extremely small, and it is a standard assumption in related works to reach $O(1/n^2)$ bounds, e.g., Remark 2 in (Zhang et al. 2017), Theorem 3 in (Xu and Zeevi, 2024), Corollary 5 in (Lei and Ying, 2020) and Theorem 22 in (Li and Liu, 2021) and so on. To see this more clear, we check the following mean estimation example.
>
> If we denote the observed data as $X_1, X_2, \ldots, X_n$ drawn from an unknown distribution, we assume that this distribution is bounded (for example, any random variable $X$ that follows this distribution satisfies $a \leq X \leq b$).
>
> To estimate the mean of the random variable, our objective function is defined as $f(m; X) = (X - m)^2$, which yields the usual least squares estimator. Using the definition of M-estimators, our goal becomes:
>
> $$
> \hat{m} = \arg\min_m \frac{1}{n} \sum_{i=1}^{n} (X_i - m)^2.
> $$
>
> We can easily verify that $f(m; X)$ satisfies the following properties: it is $4(b-a)$-Lipschitz, $2$-strongly convex, and $2$-smooth with respect to $m$. According to Theorem 4, we know that the mean estimated by this method, which minimizes the least squares loss (or the second central moment), will converge to
>
> $$
> \frac{\mathbb{V}[X] \log{(1/\delta)}}{n} + \frac{\log^2 n \log^2(1/\delta)}{n^2}
> $$
>
> as the sample size $n$ increases, where $\mathbb{V}[X]$ is the variance of the distribution. It’s worth noting that since this example pertains specifically to estimating the mean of a random variable, we have no additional parameters involved. Therefore, in this case, $F(w^{\ast}
> )$ in Theorem 5, which represents the variance of the distribution, can be very small if $\mathbb{V}[X]$ is small.
>
>
>
> > Question: Comment: The authors should specify what you mean by PGD. I thought it was GD but I have to find the original reference to know it was projected GD.
>
> Answer: At line 31, we explained that PGD stands for Projected Gradient Descent upon its first appearance. At the beginning of Section 4.2, we also provided a detailed description of the Projected Gradient Descent iterative algorithm. We appreciate your feedback and will optimize this in the final version to enhance the paper's readability.
>
>
>
>
> ## We’re happy to address any further questions you may have.
>
> Reference
>
> L. Zhang and Z.-H. Zhou. Stochastic approximation of smooth and strongly convex functions: Beyond the $O(1/t)$ convergence rate. COLT 2019.
>
> L. Zhang, T. Yang, and R. Jin. Empirical risk minimization for stochastic convex optimization: $O(1/n)$-and $O(1/n^2)$-type of risk bounds. COLT 2017.
>
> O. Bousquet and A. Elisseeff. Stability and generalization. JMLR 2002.
>
> Y. Xu and A. Zeevi. Towards optimal problem dependent generalization error bounds in statistical learning theory. Mathematics of Operations Research 2024.
>
> S. Li and Y. Liu. Improved learning rates for stochastic optimization: Two theoretical viewpoints. 2021
>
> Y. Klochkov, and N. Zhivotovskiy. Stability and deviation optimal risk bounds with convergence rate $O(1/n)$. NeurIPS 2021.
>
>
> Y. Lei and Y. Ying. Fine-grained analysis of stability and generalization for stochastic gradient descent. ICML 2020.

---

> > ### Comment · Reviewer_t9CR · 2025-08-05
> >
> > Thanks for your rebuttal. I updated my review & increased the rating.

---

> > > ### Author Response · Authors · 2025-08-07
> > > **Thank you for your high evaluation of our submission!**
> > >
> > > Thank you for your valuable comments and engaging discussion. We appreciate the time and insight you've provided to strengthen our manuscript.
> > >
> > > We are pleased to have responded to your points and addressed your concerns. We will carefully incorporate your feedback and suggestions into the manuscript during revision.
> > >
> > > We remain available until the end of the discussion phase for any further questions or clarifications needed.
> > >
> > > Sincerely,
> > > The Authors

---

### Decision · Program_Chairs · 2025-09-17

**Decision:**

Accept (poster)

**Comment:**

The paper proves fast rates for strongly convex, smooth and Lipschitz losses via stability. Specifically rates of order 1/n^2.

The reviewers agree that the results are strong. While the conditions are restrictive the paper does provide a strong theoretical push. I recommend an accept.